# High-resolution remote sensing and machine-learning-based upscaling of methane fluxes: a case study in the Western Canadian tundra

Kseniia Ivanova[1], Anna-Maria Virkkala[2], Victor Brovkin[3], Tobias Stacke[3], Barbara Widhalm[4], Annett Bartsch[4], Carolina Voigt[5,6,7], Oliver Sonnentag[7], and Mathias Göckede[1]

[1] Department for Biogeochemical Signals, Max Planck Institute for Biogeochemistry, Jena, Germany
[2] Woodwell Climate Research Centre, Falmouth, MA, USA
[3] Department Climate Dynamics, Max Planck Institute for Meteorology, Hamburg, Germany
[4] b.geos, Industriestrasse 1, 2100 Korneuburg, Austria
[5] Permafrost Research Section, Alfred Wegener Institute Helmholtz Center for Polar and Marine Research, Potsdam, Germany
[6] Institute of Soil Science, University of Hamburg, Hamburg, Germany
[7] Université de Montréal, Montréal, Québec, Canada

*Correspondence to*: Kseniia Ivanova (kivanova@bgc-jena.mpg.de)

**Abstract.** Arctic methane ($CH_4$) budgets are uncertain because field measurements often capture only fragments of the wet-to-dry gradient that control tundra $CH_4$ fluxes. Wet hotspots are over-represented, while dry, net-sink sites are under-sampled. We paired over 13,000 chamber flux measurements during peak growing season in July (2019-2024) from Trail Valley Creek in the western Canadian Arctic with co-registered remotely sensed predictor variables to test how spatial resolution (1 m vs. 10 m) and choice of machine-learning algorithm shape upscaled $CH_4$ flux maps over our 3.1 km$^2$ study domain. Four algorithms for $CH_4$ flux scaling (Random Forest (RF), Gradient Boosting Machine (GBM), Generalised Additive Model (GAM), and Support Vector Regression (SVR)) were tuned using the same stack of multispectral indices, terrain derivatives and a six-class landscape classification. Tree-based models such as RF and GBM offered the best balance of 10-fold cross-validated R$^2$ ($\leq 0.75$) and errors, so RF and GBM were used in a subsequent step for upscaling to the study area. With 1 m resolution, GBM captured the full range of microtopographic extremes and predicted a mean July flux of 99 mg $CH_4$ m$^{-2}$ month$^{-1}$. In contrast, RF, which smoothed local extremes, yielded an average flux of 519 mg $CH_4$ m$^{-2}$ month$^{-1}$. The disagreement between flux estimates using GBM and RF correlated mainly with the Normalized Difference Water Index (NDWI), a moisture proxy, and was most pronounced in waterlogged, low-lying areas. Aggregating predictors to 10 m averaged the sharp metre-scale flux highs in hollows and lows on ridges, narrowing the GBM-RF difference to ~75 mg $CH_4$ m$^{-2}$ month$^{-1}$ while broadening the overall flux distribution with more intermediate values. At 1 m, microtopography was the main driver. At 10 m, moisture proxies explained about half of the variance. Our results demonstrate that: (i) metre predictors are indispensable for capturing the wet-dry microtopography and its $CH_4$ signals, (ii) upscaling algorithm selection strongly controls prediction spread and uncertainty once that microrelief is resolved, and (iii) coarser grids smooth local microtopographic details, resulting in flattened $CH_4$ flux peaks and wider distribution. At 10 m, however, flux estimates became more consistent between models and better represented broad moisture-driven patterns, suggesting improved generalisability despite some loss of detail. All factors combined lead to potentially large differences in scaled $CH_4$ flux budgets, calling for a careful selection of scaling approaches, spatial predictor layers (e.g., vegetation, moisture, topography), and grid resolution. Future work should couple ultra-high-resolution imagery with temporally dynamic indices to reduce upscaling bias along Arctic wetness gradients.

## 1 Introduction

The Arctic is warming nearly four times faster than the global average due to Arctic amplification feedbacks (Previdi et al., 2021; AMAP, 2021; Rantanen et al., 2022; Ballinger et al., 2020). This rapid warming is of particular concern due to the substantial quantities of organic carbon stored in wetland ecosystems of the circumpolar permafrost region (Hugelius et al., 2014; Schuur et al., 2015; Turetsky et al., 2020; Olefeldt et al., 2016). Thaw exposure may mobilize part of the previously frozen carbon as methane ($CH_4$) (Ward et al., 2024), a greenhouse gas 28-34 times more potent than $CO_2$ over 100 years (Koven et al., 2011; Etminan et al., 2016; Nisbet et al., 2019; Saunois et al., 2020). Rising temperatures, therefore, risk to trigger positive feedback

in which permafrost degradation elevates CH$_4$ emissions, further intensifying warming (Schuur et al., 2015; Walter Anthony et al., 2018; Turetsky et al., 2020; Natali et al., 2021).

High-resolution CH$_4$ flux measurements in tundra ecosystems remain sparse even during the growing season due to the Arctic's remoteness, harsh climate, and logistical challenges (e.g., lengthy travel times, high fieldwork costs, sparse infrastructure, and challenging equipment maintenance), which limits the number of long-term monitoring sites (Delwiche et al., 2021). The primary tools for plot- to ecosystem scale CH$_4$ flux observations are flux chambers (Subke et al., 2021) and eddy covariance techniques, respectively (Matthes et al., 2014; Baldocchi, 2003); however, the time window to conduct growing season chamber campaigns

is usually limited to a few months between June and September, and locations in the Arctic featuring eddy covariance towers are few (Vogt et al., 2025). As a consequence, most synthesis studies aiming at constraining CH$_4$ budgets in the high northern latitudes must rely on a limited database biased toward high-emitting sites near research stations and often overlooking areas with net CH$_4$ uptake (Mastepanov et al., 2013; Varner et al., 2021; Kuhn et al., 2021; Voigt et al., 2023c). Most tundra chamber campaigns collect data only for short intervals, typically from a single day up to a few weeks during the growing season, and

many are conducted in just one growing season without repeated multi-year sampling or covering winter fluxes, which limits their value for model benchmarking and interannual analysis (Varner et al., 2021; Kuhn et al., 2021; Räsänen et al., 2021; Mastepanov et al., 2013; Treat et al., 2018).

    Even where flux data exist, CH$_4$ fluxes can shift within metres because the relative position and seasonal movement of the water table and the frost table create mosaics of anoxic (CH$_4$ - producing) and oxic (CH$_4$ - oxidising) soil (Frolking et al., 2011). These

redox contrasts are further modulated by microtopography, plant functional type, and surface moisture (Mastepanov et al., 2013; Pirk et al., 2015; Kwon et al., 2017; Olefeldt et al., 2021). Because the water table and frost table rarely coincide at the same depth across tundra microtopography, neighbouring microsites can experience very different oxic–anoxic conditions. Across the Arctic tundra, surface types range from water-saturated zones, such as sedge fens, polygon centres, troughs and thaw slumps, to better-drained features like hummocky ridges, palsas and gravelly uplands. These elements cover the entire CH$_4$ flux range, with

microtopographically lower, wetter zones acting as strong sources and microtopographically elevated, better-aerated zones often functioning as net sinks (Räsänen et al., 2021; Bao et al., 2021; Yuan et al., 2024). Such small-scale heterogeneity frequently occurs within a single 10 m pixel, so coarse maps or remote-sensing data products can combine zones of strong CH$_4$ emission and neighbouring areas that act as net CH$_4$ sinks (Knox et al., 2019; Treat et al., 2018). Without spatially explicit methods that resolve this fine-scale heterogeneity, upscaling can introduce systematic biases. It may overestimate CH$_4$ emissions when dry

areas that act as sinks are overlooked or underestimate them when narrow wet trenches surrounding dry patches are missed (Räsänen et al., 2021; Treat et al., 2018).

    Ultra-high-resolution (<1-2 m) imagery from drones or commercial satellites can directly resolve fine-scale vegetation patterns and microtopographic features (e.g., hummocks and hollows) in heterogeneous tundra landscapes, for example mapping plant communities on dry polygon rims versus wet sedge hollows and other microrelief features that correspond to CH$_4$ "coldspots"

and "hotspots", respectively. Studies using sub-metre to metre-scale imagery and plot-based observations have shown that fine spatial resolution is essential to capture local flux heterogeneity and microtopographic controls (Lehmann et al., 2016; Becker et al., 2008; Ström et al., 2005; Ludwig et al., 2024; Davidson et al., 2017). However, working with spatially ultra-high-resolution data presents significant challenges. The acquisition and processing of sub-metre or metre imagery through drones or advanced satellites and LiDAR are both costly and labour-intensive; such datasets are rarely available as dense, multi-date image stacks

and cannot be easily collected over large areas (Scheller et al., 2022; Karim et al., 2024; Anderson & Gaston, 2013). Moreover, ultra-high resolution can introduce noise from small-scale elevation artefacts and micro-relief features that do not represent real hydrological connectivity, and thus may not lead to a better representation of environmental conditions (Riihimäki et al., 2021). By contrast, high resolution (~10 m) predictors such as Sentinel-2 multispectral imagery and ArcticDEM terrain products are freely available and cover the entire Arctic with regular revisits with a standardised approach (Drusch et al., 2012; Porter et al.,

2023). However, the coarse 10 m resolution has a clear disadvantage because individual microtopographic features (e.g.,

hummocks, hollows) and landforms (e.g., dry palsas, wet trenches, etc.) that control small-scale variability in $CH_4$ fluxes are aggregated into single pixels, blurring the fine-scale patterns of emission and uptake (Räsänen & Virtanen, 2019).

Data-driven approaches, including the machine-learning (ML) algorithms Random Forest (RF), Gradient Boosting Machine (GBM), and Support Vector Regression (SVR), as well as the semi-parametric statistical model Generalised Additive Model (GAM), can integrate predictors derived from remote sensing products with flux measurements to upscale $CH_4$ from plot- and ecosystem- to landscape scales (Knox et al., 2021; Yuan et al., 2024; Chen et al., 2024; Zhang et al., 2020, Ying et al. 2025). Tree ensembles (RF, GBM) are particularly well suited for capturing complex interactions and handle multicollinearity, while GAMs have the advantage of yielding interpretable smooth functions, and SVR excels with limited nonlinear data (Wood, 2017; Smola & Schölkopf, 2004; Zhang et al., 2019). Model choice, predictor resolution and limited training data still generate large spreads in upscaled Arctic tundra $CH_4$ fluxes, with ensemble estimates differing by roughly 25-50 % of the mean depending on the study (Peltola et al., 2019; McNicol et al., 2023; Chen et al., 2024; Räsänen et al., 2021). Quantifying and reducing these uncertainties are essential for robust $CH_4$ budgets.

Here, we address these methodological challenges in a study aiming at upscaling $CH_4$ fluxes in a heterogeneous tundra landscape in the western Canadian Arctic by pairing >13,000 peak growing season (July) chamber measurements collected over five years with matched 1 m and 10 m remote sensing predictors and training three machine-learning algorithms (RF, GBM, SVR) and one semi-parametric statistical model (GAM). Our overarching aim is to reduce uncertainties in peak-season (July) $CH_4$ budgets for the 3.1 $km^2$ heterogeneous tundra around the Trail Valley Creek Research Station. We address this aim through four specific questions:

- o Which remotely sensed vegetation, moisture, and topographic characteristics best explain July $CH_4$ fluxes across a wet-to-dry micro-site gradient?
- o Does replacing freely available 10 m data (Sentinel-2, ArcticDEM) with metre imagery from drones and airborne lidar lead to a detectable improvement in prediction accuracy and spatial detail?
- o How do the four modelling approaches differ in predicted net flux magnitudes and spatial patterns?
- o How do model choice, grid resolution, and their interaction shape the spatial patterns and uncertainty of our upscaled $CH_4$ flux maps?

Optimising a data-driven upscaling approach based on these questions allows us to produce July $CH_4$ flux maps with pixel-level uncertainty, improving peak-season emission estimates and guiding where additional measurements or higher-resolution imagery would most reduce prediction error.

## 2 Materials and Methods

### 2.1 Study site

The study site is the undulating tundra landscape of the Trail Valley Creek (TVC) Research Station, about 55 km north of the town of Inuvik, NT, in the western Canadian Arctic east of the Mackenzie River Delta (Fig. 1). TVC lies in the Southern Arctic ecozone and contains continuous permafrost, with thickness ranging from 100 to 150 m (Marsh et al., 2008). Our analyses focus on a ~3.1 $km^2$ section of this 57 $km^2$ basin with elevations ranging from 41 to 102 m a.s.l. The 1991 - 2020 climate normals for Inuvik are a mean annual air temperature of –7 °C, mean annual precipitation of ~250 mm, and a frost-free period (the interval with minimum air temperatures above 0 °C) of roughly 78 days (Environment and Climate Change Canada, 2024). The soils are classified as organic cryosols, with an upper peat horizon approximately 0.2-0.5 m thick overlying mineral silty-clay subsoil (Petrone et al., 2000). The vegetation at TVC is highly diverse, reflecting the microtopography and moisture gradients. Isolated patches of white and black spruce (*Picea glauca*, P. *mariana*) occur in valley bottoms and on slopes. Tall shrub tundra, dominated by green alder (*Alnus alnobetula*) and featuring scattered willows and dwarf birch, can be found on hill slopes and alongside streams. Riparian zones feature dense willow thickets reaching up to 2 metres in height. Upland areas support dwarf shrub tundra

with dense stands of dwarf birch (*Betula glandulosa*), Labrador tea (*Ledum palustre*) and mountain cranberry (*Vaccinium vitis-idaea*), interspersed with mosses and lichens. Flat, poorly drained areas are dominated by tussock-forming sedges (*Eriophorum* and *Carex*), alongside moss and scattered shrubs. Exposed uplands and polygon rims are covered by lichen mats and low dwarf

shrubs. Mosses, especially *Sphagnum* and *Polytrichum* species, are prevalent in wetter microhabitats. Snow depth and winter soil temperatures are highest in the tall shrub and tussock zones and lowest in the lichen tundra (Grünberg et al., 2020; Marsh et al., 2010). Although TVC represents a single site, its strong microtopographic and vegetation heterogeneity reflects the wet-dry gradients typical of Arctic continuous-permafrost lowlands. Similar mosaics of sedge wetlands, dwarf-shrub uplands, and lichen tundra occur across large parts of the western Canadian Arctic and other low-relief tundra landscapes, suggesting that the scale

effects we document are broadly transferable.

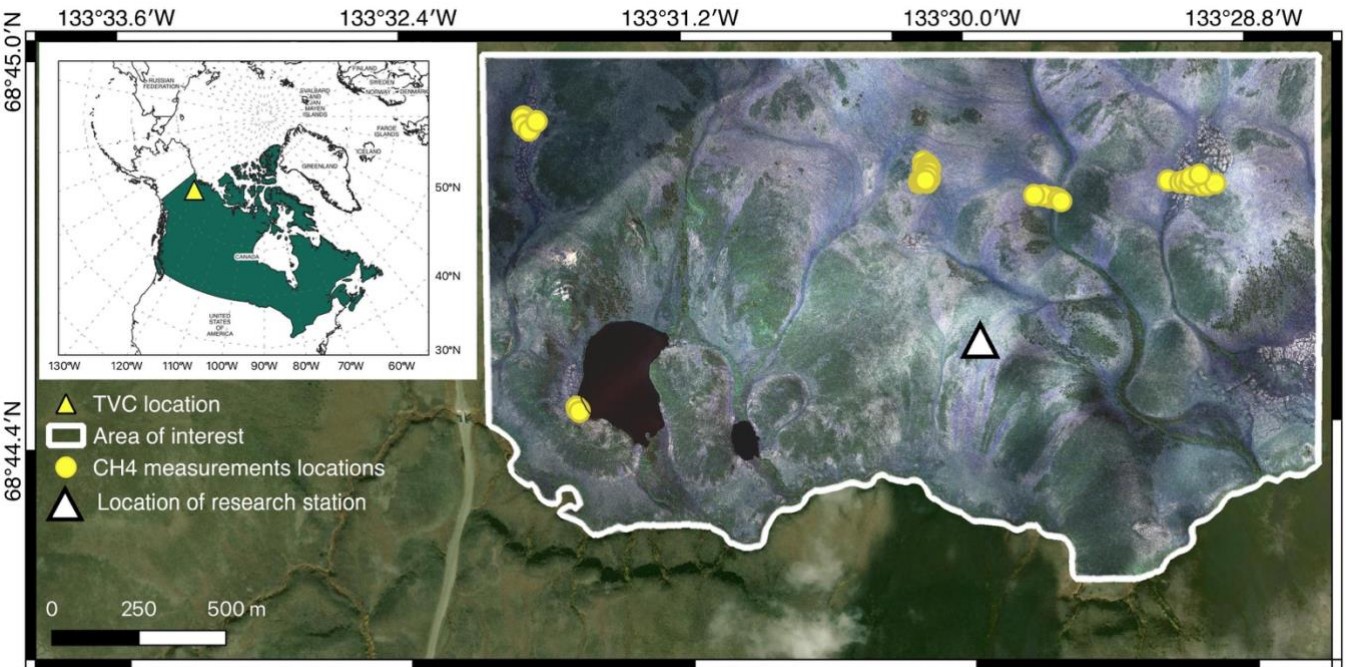

**Figure 1. Map of the study area showing the location of the area of interest (outlined with white polygon), with CH4 flux measurement locations marked with yellow circles. The inset map in the upper left corner highlights the region in which the Trail Valley Creek (TVC) research station is located (marked with a yellow triangle on the overview map and a white triangle on the detailed map).**
**Background satellite imagery sources: © Maxar 2025, provided by Esri, acquired on 12 July 2024. Area of interest aerial imagery: Rettelbach et al., 2024.**

## 2.2 Data sources

This study combines field-based $CH_4$ flux measurements with remotely sensed and meteorological data to build and evaluate spatially explicit models of $CH_4$ exchange. The chamber flux data provide the response variable for model training, while the
meteorological records include air temperature (AT), photosynthetically active radiation (PAR), and thawing degree days (TDD) as dynamic atmospheric drivers. Remotely sensing datasets supply spatial predictors describing vegetation, surface moisture, terrain structure, and landscape classification at two spatial resolutions (1 m and 10 m). The resulting predictor stacks were then used to train and compare the four modelling approaches described in Sect. 2.3.

### 2.2.1 CH4 flux data

We used a combination of continuous and campaign-based $CH_4$ flux measurements to capture spatial and temporal variability in $CH_4$. The dataset includes previously published automated chamber observations made in 2019 and 2021 (Voigt et al., 2023a), and campaign-based manual chamber observations made in 2019 (Voigt et al., 2023b) and in 2022 to 2024 (Ivanova et al., 2025). Manual chamber measurements from 2022 to 2024 were collected as part of this study. The main measurement protocols, chamber specifications, instrumentation, and flux calculation methods for each campaign are summarized in Table 1.

**Table 1. Summary of CH₄ flux measurement protocols and instrumentation used at TVC, 2019–2024.**

| Year | 2019, 2021 | 2019 | 2022–2024 |
|---|---|---|---|
| **Method** | Automated | Manual | Manual |
| **Number of microsites** | 18 | 13 | 37 |
| **Chamber size and shape** | 30-45 L, hemispherical | 17 L, cylindrical | 17 L, cylindrical |
| **Gas analyzer** | Los Gatos Research Enhanced Performance Greenhouse Gas Analyser (Rackmount GGA-24EP 911-0010, Los Gatos) | Picarro G4301 GasScouter (Picarro, Inc., Santa Clara, CA, USA) | LI-COR LI-7810 Trace Gas Analyzer (LI-COR Biosciences, Lincoln, NE, USA) |
| **Measurement frequency** | 1 Hz | 1 Hz | 1 Hz |
| **Enclosure time** | 3 min | 5 min | 2–4 min |
| **Flux calculation method** | Linear regression (default); exponential fit for large fluxes | Linear or nonlinear regression with the Math Works Inc., Natick, MA, USA) | Linear regression with bootstrapping (R) |
| **Reference** | Voigt et al., 2023a | Voigt et al., 2023b | Ivanova & Göckede, 2025 |

The complete dataset included 13,384 CH₄ flux measurements collected between 1 July and 31 July (2019-2024) under both light and dark conditions. Our chamber measurements cover the spatial heterogeneity of the ~3.1 km² study area, ensuring representation of key CH₄ controlling gradients.

Flux measurements were collected across the full range of microtopographic and vegetation conditions within the AOI. Observations were distributed across tussock tundra, dwarf shrubs, lichen-dominated uplands, and sedge wetlands at both spatial resolutions. The sampling distribution closely matched the mapped area fractions of these classes in the AOI (Fig. S1), confirming robust ecological representativeness. Detailed percentages for both map area and flux sampling are provided in Fig. S1. Repeated measurements under different meteorological conditions also provide independent temporal variability for model training. On average, each microsite was measured 50-450 times depending on year and instrument type, resulting in a total of 13,384 individual chamber observations across 68 unique locations (microsites). Of these, 1,093 fluxes were measured manually using closed chambers, while 12,291 were collected using an automated chamber system (Fig. 1). For manual chamber measurements, collars were installed in the soil prior to the actual measurements to ensure a tight sealing of the volume enclosed by the chamber hood. For these locations, no boardwalks were installed to access the site, but activities close to the site before and during the measurements itself were avoided as much as possible to keep potential disturbances to a minimum. These narrow boardwalks did not overlap with the chamber footprints, and therefore should not affect remote sensing data at 1m resolution, while an impact on 10m pixels is considered minimal. For each flux measurement, ancillary data recorded include coordinates, PAR (measured as photosynthetic photon flux density (PPFD; μmol m⁻² s⁻¹), air temperature, land cover type, and time of day (when available).

### 2.2.2 Climatic data

AT data were obtained from the Trail Valley Creek meteorological station operated by Environment and Climate Change Canada – Meteorological Service of Canada (ECCC, 2024). The station is located within the study area at 68°44′46.8″ N, 133°30′06.4″ W, at an elevation of 85 m a.s.l. (Climate ID: 220N005; WMO ID: 71683; TC ID: XTV). The original data were recorded at hourly resolution and were downsampled to 3-hour intervals to match the temporal resolution of the model predictions. PAR data were obtained from the NASA POWER dataset (Langley Research Center, 2024) at a spatial resolution of 1 km. These data provided temporally dynamic inputs for model training and prediction.

### 2.2.3 Remotely sensed data

We assembled two separate but equivalent predictor stacks, one with a cell size of 1 m and one with 10 m. Both cover the same area of interest (AOI, Fig. 1), use the same map projection, and pass through the same preprocessing workflow (Ivanova et al., 2025). The AOI was delineated along natural drainage lines on three sides, and the Inuvik-Tuktoyaktuk Highway along the western boundary. An image stack refers to a set of co-registered raster layers (multispectral indices and terrain derivatives) that share the same grid and extent. To facilitate comparison between datasets of different spatial resolutions, we summarized all predictors in Table A1. It lists each variable with its data source, spatial resolution (1 m, 10 m, or non-spatial), and whether it is static or dynamic. Variables derived from UAV imagery are used at 1 m resolution, while Sentinel-2 and ArcticDEM products are used at 10 m.

The 1 m stack is based on the RGB + NIR drone orthomosaic captured on 22 August 2018 by Rettelbach et al. (2024) and the 1 m LiDAR-derived digital terrain model (DTM) from Lange et al. (2021). From these layers, we derived the Normalised Difference Vegetation Index (NDVI; Rouse et al., 1974) and the Normalised Difference Water Index (NDWI; Gao, 1996; McFeeters, 1996) as proxies for biomass and surface moisture, respectively. Topographic derivatives including slope, aspect, the Topographic Position Index (TPI, 30 m window), and the Topographic Wetness Index (TWI) were calculated with Whitebox Tools (Lindsay, 2016). A 30 m neighbourhood was used for TPI, as this scale best captured local elevation contrasts typical of heterogeneous microtopography.

The 10 m stack contains the same set of variables but at coarser spatial resolution. It combines multispectral information from Sentinel-2 Level-2A scenes collected between 2015 and 2024 (Copernicus, 2024) with topographic derivatives derived from the 2 m ArcticDEM, resampled to 10 m to match the Sentinel grid. Cloud-, shadow-, and snow-masked for AOI Sentinel-2 Level-2A scenes from July-August 2018 (n = 6 cloud-free scenes) were composited in Google Earth Engine using the mean to align with the 2018 drone campaign (Gorelick et al., 2017). For the time-specific analysis, NDVI and NDWI were extracted from the nearest cloud-free scene within ±10 days of each chamber measurement, with no temporal averaging and only cloud-free pixels accepted. NDVI and NDWI were extracted from this composite, and the same set of terrain derivatives (slope, aspect, TPI, TWI) was computed for consistency. A complete overview of all predictor variables, including data descriptions, resolution, temporal variability, and references, is provided in Appendix Table A1. To link chamber measurements with remote sensing inputs, predictor values were extracted directly from the raster cell covering the chamber footprint, without spatial buffering. No spatial averaging or neighbourhood smoothing was applied to the pixel values at extraction. All chamber measurements were kept as individual records, even when multiple chambers or repeated measurements fell within the same 1 m or 10 m grid cell, to preserve sub-pixel heterogeneity in vegetation and soil conditions.

In a separate workflow, we used multispectral, terrain, and texture features to produce a site-specific landscape classification map at both 1 m and 10 m resolution using a Random Forest approach (Breiman, 2001). The 1 m dataset was derived from RGB + NIR orthomosaic drone imagery collected by drone on 22 August 2018 (Rettelbach et al., 2024) and a co-registered 1 m LiDAR-based digital terrain model (Lange et al., 2021). The 10 m dataset was based on Sentinel-2 multispectral imagery and ArcticDEM-derived terrain parameters, representing the same area of interest. Six landscape classes were defined following Grünberg et al. (2020): Water, Lichens, Tussock, Dwarf Shrubs, Tall Shrubs + Trees, and Sedges.

Training and validation points (n = 140 in total) were manually delineated from the drone orthomosaic. Eighty percent of the points were used for model training and 20 % for accuracy assessment. The same training polygons were used for both the 1 m and 10 m classifications in terms of geographic location and class label, while predictor values were extracted from the respective remote-sensing datasets (drone + LiDAR for 1 m; Sentinel-2 + ArcticDEM for 10 m). This approach ensured that the two classifications were comparable while reflecting the characteristics of their respective input data. Because the spatial resolution and input data differ, the resulting landscape maps do not show identical boundaries or class proportions, but instead reflect the surface characteristics captured at each scale. Both maps contained the same six land-cover classes. However, for the 1 m model training, the Tall shrubs + trees class was merged with Dwarf shrubs because no chamber flux measurements overlapped that

class. Thus, five classes were used for flux modelling at 1 m, whereas all six were retained at 10 m. Water pixels were masked prior to classification using a threshold of NDWI > 0 and manually checked against the drone orthomosaic to ensure the exclusion of ponds and streams. All subsequent statistical analyses were restricted to terrestrial classes. A detailed description of the classification workflow, feature set, and accuracy assessment is provided in Text A1 and Table A2.

For broader application, the 10 m predictor stack is directly reproducible across the Arctic (Sentinel-2 Level-2A + ArcticDEM). In contrast, the 1 m stack depends on site-specific drone orthomosaics and LiDAR, which limits immediate circumpolar scaling but is valuable for local calibration and bias assessment.

  In addition, we also explored the potential of two datasets that are particularly relevant for Arctic-scale applications. The Circumarctic Land cover Units (CALU) (Bartsch et al., 2024) provides a 10 m classification of vegetation physiognomy and soil

moisture regimes across the circumpolar Arctic tundra. The data product is based on the fusion of Sentinel-1 and Sentinel-2 imagery and was calibrated using over 3,500 field samples of soil and vegetation properties. One of the key strengths of CALU is that it captures spatial gradients in surface wetness while using a consistent classification scheme across all Arctic regions. This makes it possible to directly compare classes between distant sites across the Arctic, which is rarely achievable with site-specific classifications. 20 of 23 land cover units are found across the AOI, but only 5 of those were covered by $CH_4$

measurements. The complete legend of CALU classes used in this study, including definitions, their occurrence within the AOI, and whether $CH_4$ flux measurements are available for each class, is presented in Table A3. Additionally, we considered a radar interferometric (InSAR) dataset derived from Sentinel-1 data for 2018 - 2023 (Widhalm et al. 2025), which captures seasonal ground subsidence rates in thawing degree days domain associated with thaw table (the uppermost soil that freezes and thaws each year). The magnitude of the subsidence rates reflects soil moisture gradients (Widhalm et al. 2025).

Finally, we assessed the benefit of incorporating time-specific spectral indices (NDVI and NDWI) extracted from Sentinel-2 scenes close to each chamber measurement. We compared the effect of using these time-matched indices versus a seasonal composite (July – August 2018) to test whether short-term variability in vegetation and moisture status improves model skill. Although this approach relies on satellite scenes taken within a limited time window and may not align perfectly with the exact in situ measurement date, it still offers a more detailed representation of changes in surface conditions than seasonal averages.

All four additional predictors (CALU, InSAR, and temporally dynamic NDVI and NDWI) were tested in separate model runs to assess how much they improved predictive performance ($R^2$, RMSE) when added to the main predictor set. These sensitivity analyses allowed us to evaluate their explanatory value without altering the resolution comparison, as each of these variables is available only at a single spatial scale.

  To specifically evaluate how spatial resolution influences model performance and predicted $CH_4$ fluxes, we designed the analysis

so that the only factor differing between the two datasets was grid size (1 m vs. 10 m). All other parameters (predictors, preprocessing steps, algorithms, and training data) were kept identical. This approach allowed us to isolate the effect of scale from other potential sources of variation. As shown in Fig. 2, narrow, wet features such as polygonal trenches are captured at 1 m but blended at 10 m, which alters both the NDVI and the landscape classification. Since many high $CH_4$ fluxes originate from these small wet zones, aggregation at a coarser resolution obscures their contribution. Some of the remaining model disagreement

may also be due to the limited representation of extremely wet or complex terrain in the training data, which reduces the model's generalisability.

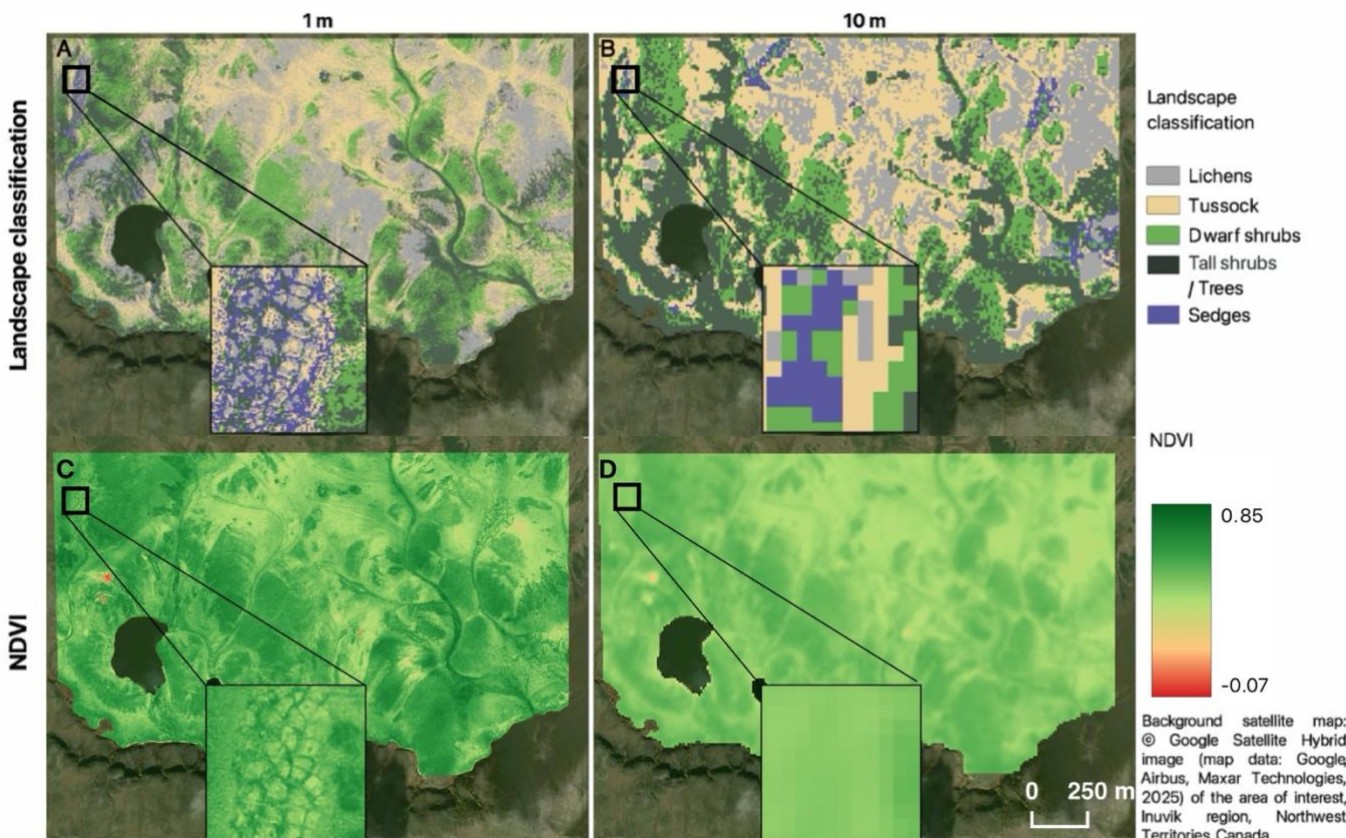

**Figure 2. Site-specific landscape classification (LC) and Normalized Difference Vegetation Index (NDVI) at two spatial resolutions: 1 m (panels A and C) and 10 m (panels B and D). Panels A and B show LC maps, while panels C and D show NDVI. Each panel includes a black-framed inset highlighting a representative polygonal mire. Narrow, waterlogged microtopographic features such as wet trenches remain distinct at 1 m resolution but blend into mixed pixels at 10 m. Background imagery: © Google Satellite Hybrid (Maxar, 2025).**

### 2.3 Statistical analyses

The statistical analysis was structured into five sequential stages: (1) data preparation, (2) model training and evaluation, (3) spatial prediction, (4) temporal aggregation and interpretation, and (5) variable importance analysis (Fig. 3). All steps were applied identically to the 1 m and 10 m datasets to enable direct comparison of model behaviour and prediction outcomes across spatial resolutions. The analysis was implemented in R 4.3.2 (R Core Team, 2024).

### 2.3.1 Data preparation

The first step consisted in the preparation of the predictor datasets to explain spatio-temporal variability in $CH_4$ fluxes. In total ten predictors were used: AT, PAR, TDD, NDVI, NDWI, slope, aspect, TPI, TWI, and a six-class landscape classification (see Appendix Table A1 for details). The three meteorological variables (AT, PAR, and TDD) were treated as spatially uniform across the ~3 km² study area, as it is covered by a single meteorological station. Their values varied only temporally, while all other predictors were spatially distributed but static over time during each model run. To assess potential multicollinearity, pairwise correlations among predictors were calculated separately for the 1 m and 10 m datasets using Spearman's rank correlation.

### 2.3.2 Model training and evaluation

Second, we evaluated four modelling families for their ability to predict $CH_4$ fluxes: random forests (RF), gradient-boosting machines (GBM), generalized additive models (GAM), and support-vector regression (SVR). RF is a ML algorithm that builds multiple decision trees on bootstrapped data. The mean of their outputs is then calculated. The averaging reduces noise and the method reports easy-to-read variable-importance scores (Breiman 2001; Prasad et al. 2006). GBM also uses trees but adds them

one after another. Each new tree learns from the errors of the current ensemble, which often reduces bias but requires careful tuning to avoid over-fitting (Friedman 2001; Elith et al. 2008). Similar RF, GBM handles mixed predictor types, outliers, missing values, and nonlinear relationships without preprocessing (Elith et al., 2008). GAM is a statistical technique that fits a smooth curve to each predictor and then combines these curves to create a composite curve. The curves demonstrate how $CH_4$ changes with each driver and provide reliable predictions beyond the training range (Hastie & Tibshirani, 1990; Wood, 2017). SVR is a ML algorithm that fits a flexible line or surface that best follows the data while allowing small errors within a defined range. It uses a mathematical function called a kernel to handle weak non-linear patterns, and is particularly effective when the dataset is small or the relationships are not strongly linear (Cortes & Vapnik 1995; Smola & Schölkopf 2004). Each model was implemented using the **caret** package in R (Kuhn 2008) for tuning and evaluation via stratified 10-fold cross-validation. We used the R-packages ***ranger*** for RF (Wright & Ziegler, 2017), ***gbm*** for GBM (Greenwell et al. 2022), ***kernlab*** for SVR (Karatzoglou et al. 2004), and ***mgcv*** for GAM (Wood, 2017). Model performance was assessed using five-fold cross-validation based on out-of-fold predictions. Three complementary metrics were used: the coefficient of determination ($R^2$), root mean square error (RMSE), and mean absolute error (MAE). $R^2$ describes how well model predictions capture the variability of observed $CH_4$ fluxes, RMSE emphasises large deviations, and MAE quantifies the average absolute difference between observed and predicted values. $R^2$ and RMSE were the main criteria for evaluating predictive performance and selecting the best model configurations, while MAE was reported as an additional indicator of absolute error, given the low mean $CH_4$ fluxes. All metrics were computed from cross-validated predictions using the ***yardstick*** package (Kuhn et al., 2025) to ensure consistent implementation across all model types.

We tuned the key parameters of RF, GBM, SVR, and GAM using five-fold cross-validation with RMSE as the evaluation metric (Text S1). For SVR, several kernel functions were tested and the radial basis function kernel provided the best performance. GAMs were fitted using thin-plate regression splines for numeric predictors and penalization of uninformative smooth terms. Multicollinearity among predictors was assessed using Variance Inflation Factors (VIF/GVIF) and GAM concurvity diagnostics, and no predictors exceeded commonly used concern thresholds (Text S2, Table S2). Therefore, the full predictor set was retained at both spatial resolutions.

In addition to the main predictor set, models were also trained with additional variables available only at 10 m spatial resolution, including CALU land cover, InSAR-derived surface subsidence, and temporally dynamic NDVI and NDWI extracted for dates closest to each $CH_4$ flux measurement. These variables were tested in separate model runs to evaluate their explanatory power but were not included in the main inter-resolution comparison, as they were unavailable at 1 m resolution and would otherwise bias scale-related analyses.

To disentangle the effects of spatial resolution and data source, we additionally aggregated the 1 m input dataset to 10 m resolution using the same workflow. Continuous predictors were averaged within each 10 m grid cell, and categorical variables (LC) were assigned based on the majority class. These aggregated data were then used to train and evaluate all models using the same hyperparameter settings and cross-validation strategy as for the main analysis.

### 2.3.3 Spatial prediction

Third, two best-performing models (RF and GBM) were applied to a complete spatial predictor stack, a multi-layer raster covering the entire study area without gaps. The stack included two types of layers. Static layers, such as NDVI, NDWI, slope, aspect, TPI 30 m, TWI, and land cover, remained unchanged throughout July. In contrast, the meteorological layers (AT, TDD, PAR) were spatially uniform but temporally dynamic. A temporal loop progressed from 1 July at 00:00 to 31 July at 23:59 in three-hour steps. At each time step, the corresponding values of AT, PAR, and TDD were inserted into their respective layers in the stack. The model then generated an instantaneous $CH_4$ flux raster in mg $CH_4$ m$^{-2}$ h$^{-1}$ using the ***terra*** package (Hijmans 2023). This routine resulted in 248 flux rasters for the whole month of July, produced per year and spatial resolution. To ensure

consistency, areas with missing input values (e.g., water bodies) were excluded from predictions. In total, 5,952 $CH_4$ flux rasters were generated (248 time steps × 2 models × 2 resolutions × 6 years).

### 2.3.4 Temporal aggregation and interpretation

Fourth, the predicted raster time series was aggregated using arithmetic operations in the *terra* package. Averaging over all time steps resulted in July mean flux maps, while summing and multiplying by three converted instantaneous rates into cumulative monthly fluxes. Six-year means and interannual variation (2019 to 2024) were calculated. To assess spatial mismatches related to scale, the 1 m predictions were aggregated to 10 m resolution, and the aggregated 1 m maps were subtracted from the 10 m maps pixel-by-pixel to compute spatial differences. In addition, differences between the two tree-based model families, RF and

GBM, were mapped to quantify structural uncertainty. To interpret these mismatches, Pearson correlations were calculated between the difference maps (resolution- or model-based) and individual predictor layers.

### 2.3.5 Variable importance analysis

Fifth, we conducted a separate variable importance analysis to identify the most influential predictors in each model. Variable importance scores were extracted from the cross-validated, hyperparameter-optimised RF and GBM models using permutation

importance (*ranger* package; Wright & Ziegler, 2017) and relative influence (*gbm* package; Greenwell et al., 2022), respectively. These scores were used to assess the consistency of predictor relevance across models and spatial resolutions. Variable importance scores for each model were normalized by dividing by the sum of all importance values within that model, resulting in relative importance values ranging from 0 to 1.

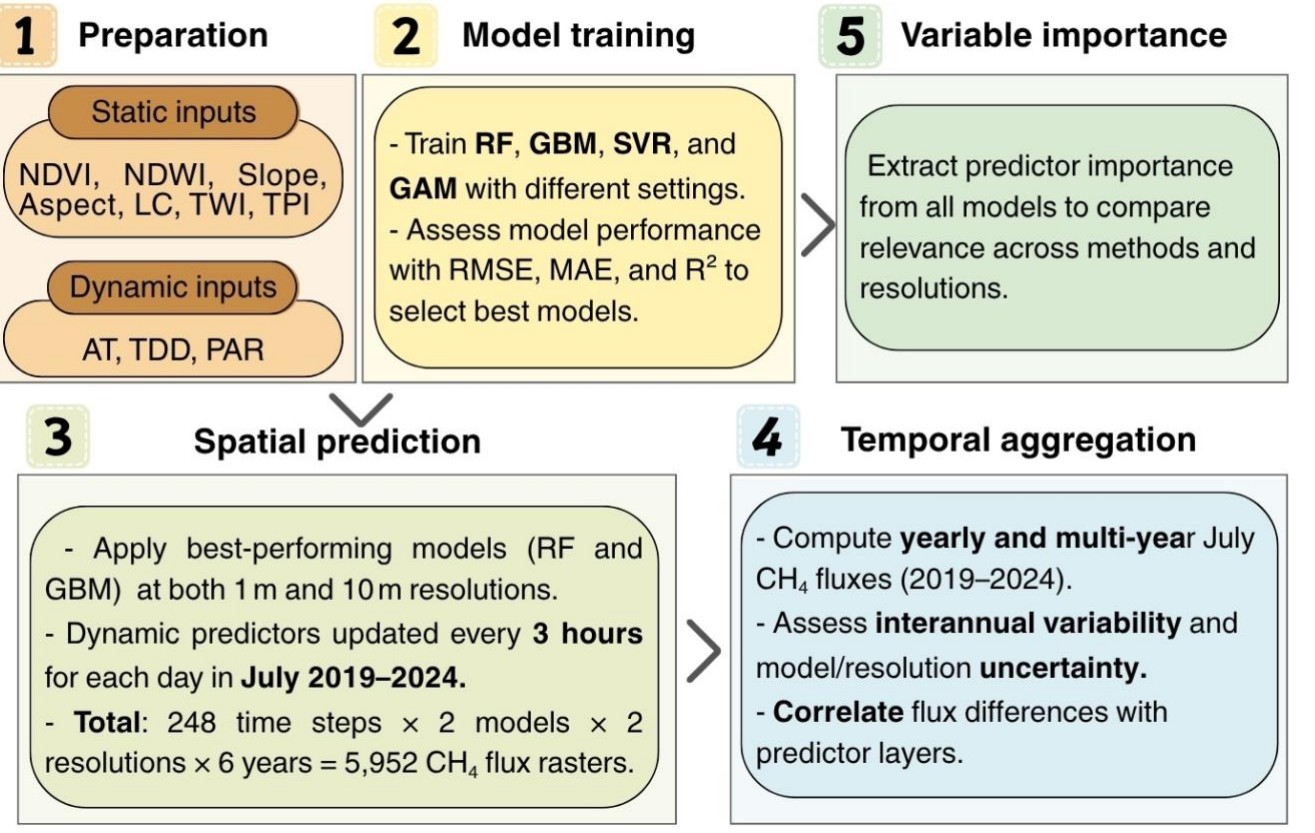

**Figure 3. Workflow for modelling and upscaling $CH_4$ fluxes in the designated study area. The analysis was performed separately for 1 m and 10 m spatial resolutions and comprised five primary stages: (1) Predictor preparation. (2) The training and tuning of models. (3) Spatial prediction. (4) Temporal aggregation and evaluation. (5) Variable importance.**

## 3 Results and Discussion

### 3.1 Correlation between observed CH₄ fluxes and single remote sensing parameters

This exploratory analysis examined how observed July $CH_4$ fluxes correlate with individual environmental variables at two spatial scales (1 m and 10 m) to identify significant controls of $CH_4$ flux and how spatial resolution affects their predictive power. Although the observed associations were generally weak, several clear patterns emerged across landscape classes and environmental gradients.

Seasonal subsidence showed the strongest positive correlation, underscoring the explanatory power of this parameter for moisture availability and related enhancements in $CH_4$ fluxes (Table 2). This is in line with observations linking InSAR-derived subsidence to elevated $CH_4$ fluxes in Arctic ecosystems (Sjögersten et al., 2023). Several moisture-related indices (NDWI, TWI, TPI) show higher correlations at 10 m than at 1 m, because 10 m aggregation smooths microtopographic noise while 1 m retains over-detailed, heterogeneous signals. This indicates that coarser resolution better captures landscape-scale hydrological gradients. This finding is supported by Ruhoff et al. (2011), who demonstrated that TWI values stabilise and become more spatially coherent at coarser resolutions, and by Riihimäki et al. (2021), who showed that TWI's ability to predict soil moisture improves when derived from coarser DEMs (e.g., 10-30 m). Conversely, the correlation with aspect weakened at 10 m, compared to 1 m resolution, likely due to the loss of microtopographic detail when pixels are aggregated, as shown previously (Schoorl et al., 2000; Vaze et al., 2010).

Temporally matched NDVI and NDWI show weaker correlation with $CH_4$ fluxes compared to static indices. The reason may be the limited effective temporal resolution of Sentinel-2: although the constellation has a nominal 5-day revisit, persistent Arctic cloud cover often stretches the cloud-free gap well beyond 10 days (Runge & Grosse, 2019), producing a temporal mismatch with chamber measurements.

**Table 2. Spearman rank-correlation coefficients (ρ) between July CH₄ flux and environmental predictors at 1 m and 10 m spatial resolution. Positive values indicate that higher predictor values coincide with higher CH₄ emissions, negative values indicate the opposite. All correlations were computed using the full dataset. For vegetation and surface wetness predictors (NDVI, NDWI), both static (July 2018) and temporally matched values (Sentinel-2 scenes within ±10 days of each chamber measurement) are shown. The column "10 m, temporal" reflects those temporally matched predictors. For predictors derived from static landscape characteristics (e.g., TWI, Slope, TPI, Subsidence), 10 m and 10 m-temporal columns are merged as they do not vary in time. Significance levels: **p < 0.01, ***p < 0.001.**

| Group | Predictor | 1 m | 10 m | 10 m, temporal |
|---|---|---|---|---|
| Vegetation | NDVI | -0.289*** | -0.295*** | -0.082** |
| Surface wetness and soil moisture | NDWI | 0.141*** | 0.24*** | -0.013 |
| | TWI | 0.027** | 0.235*** | |
| Topography | Slope | -0.187*** | -0.238*** | |
| | Aspect | 0.14*** | 0.035*** | |
| | TPI | -0.162*** | -0.327*** | |
| Ground subsidence | Cumulative seasonal subsidence | | 0.534*** | |

We examined $CH_4$ flux variation across the landscape classes and CALU units (Fig. 4A and Table B1). For example, sedge-dominated landscape classes had the highest mean $CH_4$ flux (0.87 – 0.94 mg $CH_4$ m$^{-2}$ h$^{-1}$). Elevated fluxes in these systems are likely driven by plant transport through aerenchymatous tissue during which $CH_4$ produced at depth bypasses the oxic zones,

and enhanced $CH_4$ production resulting from high plant productivity and increased substrate availability via root exudates (Olefeldt et al., 2013; Kwon et al., 2017). Tussock areas displayed the lowest flux values, with on average minor uptake of $CH_4$ (-0.02 mg $CH_4$ m$^{-2}$ h$^{-1}$). These patterns were consistent with observations by Voigt et al. (2023c).

All pairwise differences between $CH_4$ flux distributions for the 1 m and 10 m products were statistically significant (Wilcoxon rank-sum test, $p < 0.0001$). However, this result should be interpreted with caution due to the large sample sizes (even subtle

differences can appear significant). In some cases, the differences in median fluxes were small (e.g., sedges), while in others, the resolution shift results in more substantial changes (e.g., dwarf shrubs: median increased from 0.05 to 0.19 mg $CH_4$ m$^{-2}$ h$^{-1}$). In some cases, the flux sign even changed, for instance, lichen-dominated areas shift from weak uptake to weak emission. These shifts likely reflect the effects of aggregation, where coarser resolution mixes surface types or blends microsites with different flux patterns.

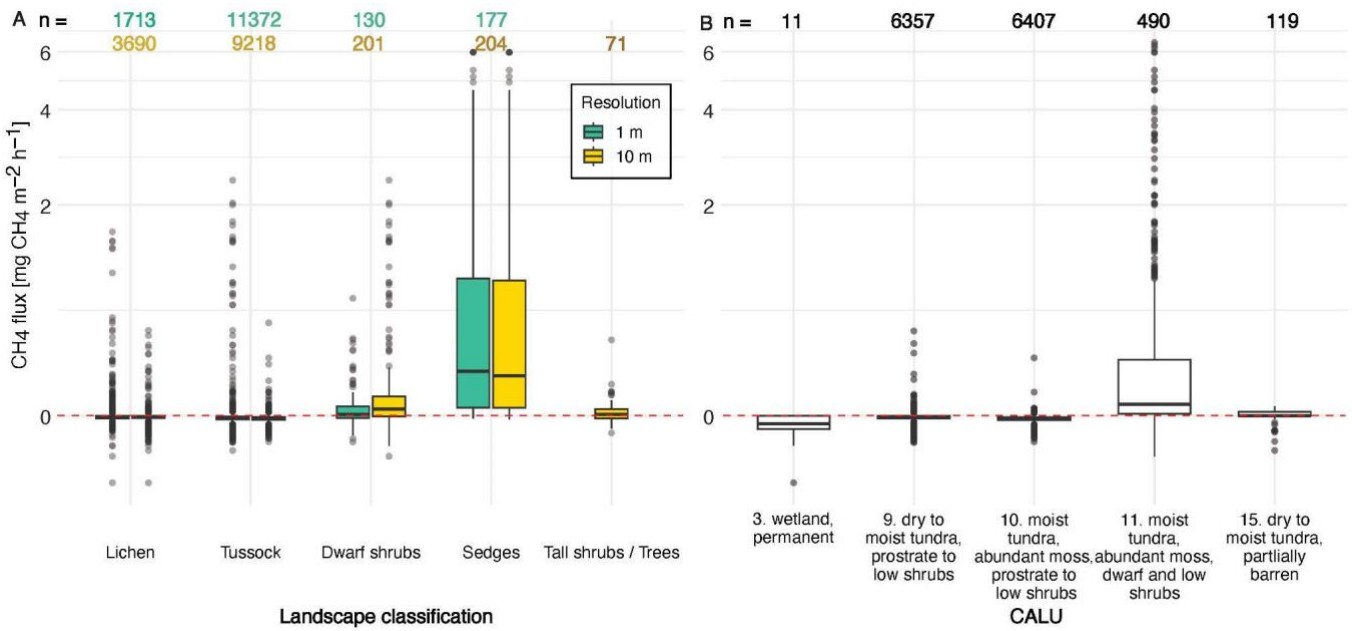

**Figure 4. Comparison of observed $CH_4$ fluxes across site-specific landscape classification at two spatial resolutions and CALU vegetation classes. Panel A: $CH_4$ fluxes across five site-specific landscape classes with existing $CH_4$ flux measurements. Measurements were aggregated separately for 1 m and 10 m spatial resolution. Panel B: $CH_4$ fluxes grouped by CALU (Circumarctic Land cover Units) classes. $CH_4$ fluxes differed significantly between most CALU classes ($p < 0.001$, pairwise Wilcoxon test), except classes 10 and**

**3 (wetlands), where no significant difference was observed ($p = 0.054$). Boxplots show the distribution of fluxes for each group. horizontal lines represent medians, boxes indicate the interquartile range, and whiskers extend to 1.5× the IQR. The red dashed lines indicate zero fluxes.**

To assess how well a pan-Arctic land-cover scheme captures $CH_4$ flux variation, we aligned our measurements with the CALU map (Fig. 4B). CALU vegetation classes differed significantly in $CH_4$ flux, except between moist moss tundra, abundant moss,

prostrate to low shrubs (class 10) and permanent wetlands (class 3) (Fig. 4B, Table B2). Within CALU classes, average $CH_4$ fluxes ranged from slight uptake in wetland class (-0.09 mg $CH_4$ m$^{-2}$ h$^{-1}$) to moderate emissions in moist tundra, abundant moss, dwarf and low shrubs (CALU 11) (0.46 mg $CH_4$ m$^{-2}$ h$^{-1}$). Unexpectedly, the permanent wetland class showed $CH_4$ uptake. This category only included one area, where dry lichen areas dominate most of the area. Moreover, the 10 m resolution of CALU likely leads to mixed pixels, where wetter spots were averaged with drier surroundings, reducing the apparent $CH_4$ emissions. In

contrast, many wet areas at our site were too small to be resolved as wetlands in CALU and were instead classified into other categories.

Overlay analysis between our site-specific landscape classification and the CALU (Fig. 5) showed that each of our landscape classes included 6-11 CALU classes (with coverage > 1 %), typically dominated by moist tundra, abundant moss, dwarf, and low shrubs (CALU 11). This reflects differences in classification approaches: CALU aimed at representing vegetation diversity

and wetness gradients across the entire Arctic (Bartsch et al., 2024), whereas the site-specific landscape classification was explicitly built for $CH_4$ flux modelling and therefore integrates fine-scale microtopography, surface-moisture patterns, and local

vegetation. A similar degree of cross-class mixing was observed for the 1 m classification (Fig. S2), indicating that these differences are primarily driven by conceptual distinctions between classification schemes rather than spatial resolution.

While both CALU and our site-specific classifications captured broad vegetation and wetness gradients, tall-shrub areas were clearly underrepresented in the flux dataset (< 1.5 % of points vs. ~20-30 % of the mapped area (Fig. S1). These zones often coincide with wetter micro-depressions and drainage areas (Grünberg et al., 2020), suggesting that the wettest conditions are not fully captured in the current sampling. Increasing coverage of these habitats would improve the robustness of flux comparisons and reduce residual variability in future models.

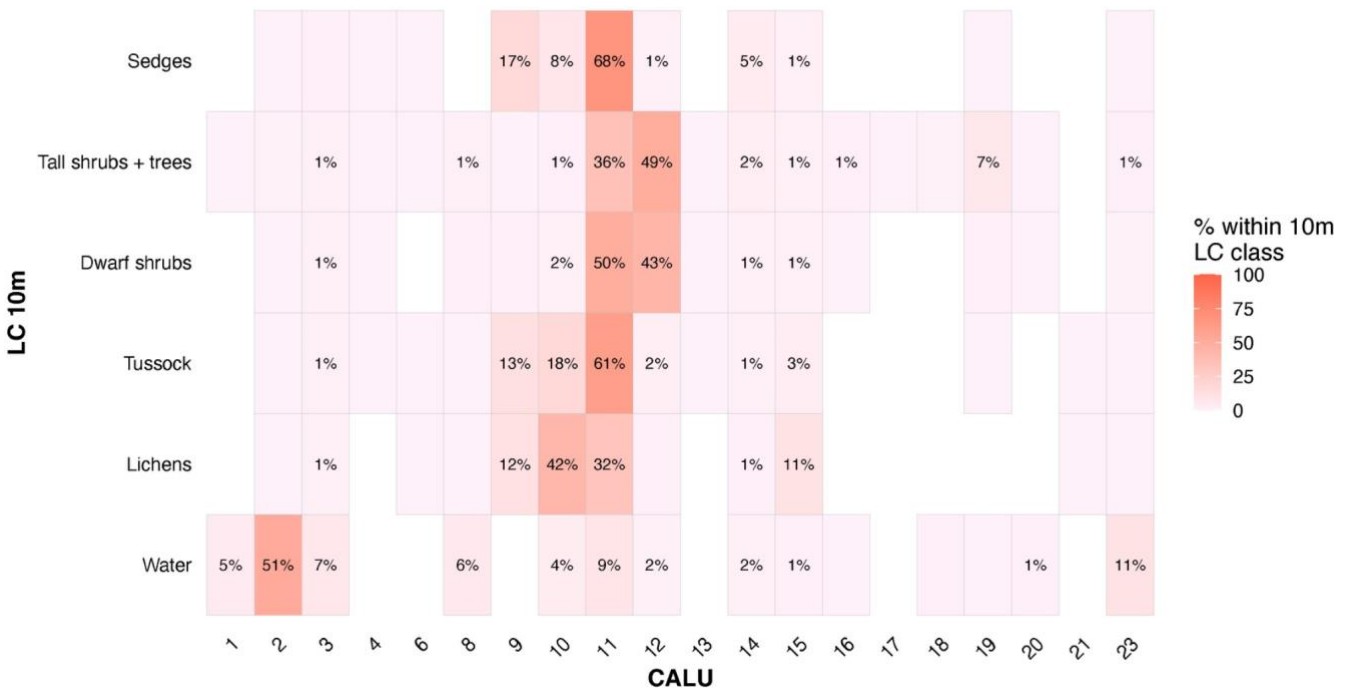

Figure 5. Pixel-wise cross-comparison between two 10 m land-cover products for the TVC study area. LC 10 m (this study): a site-specific Sentinel-2 + ArcticDEM classification built (see SI Text 1). CALU (Circumarctic Land cover Units): published pan-Arctic landcover units (full legend in Table A3). Each tile shows the fraction of pixels of a given CALU class that fall into that LC 10 m class; row totals, therefore, equal 100 %. Values ≥ 0.5 % are printed inside the tiles. Tiles that are coloured but unlabelled occur (< 0.5 %), while blank tiles indicate class pairs that do not intersect within the AOI.

However, even within each CALU or LC class, flux variance remained high, underlining that vegetation type alone cannot capture the full pattern of CH$_4$ fluxes without considering microtopography and moisture indices. Similar to the pan-Arctic synthesis by Olefeldt et al. (2013), our findings support the view that the effects of key environmental parameters on CH$_4$ flux should be considered jointly rather than independently. Additionally, soil temperature and soil moisture, key controls of CH$_4$ production and oxidation (Wille et al., 2008; Mastepanov et al., 2013), were not included as predictors in the present analysis because no high-quality gridded datasets were available that matched the spatial resolution and thematic detail required for our modelling framework. These variables are planned for integration in future model development as suitable products become available.

## 3.2 Evaluation of Model Accuracy

Our cross-validated modelling framework achieved predictive performance (R$^2$ from 0.53 to 0.87, Table 2) comparable to recent CH$_4$ upscaling studies in the Arctic-boreal region, including both chamber- (e.g., Virkkala et al., 2023; Räsänen et al., 2021) and eddy covariance-based studies (e.g., McNicol et al., 2023; Chen et al., 2024; Peltola et al., 2019; Tramontana et al., 2016). Model evaluation at 1 m resolution revealed that SVR achieved the highest R$^2$ of 0.87, indicating strong predictive power. However, this was accompanied by substantial errors (RMSE = 0.078, MAE = 0.019 of mean CH$_4$ flux), suggesting high sensitivity to skewed distributions and outliers, a known limitation of SVR when modelling non-Gaussian ecological data (Smola & Schölkopf, 2004). In contrast, RF showed both high accuracy and robustness, combining high R$^2$ with the lowest errors among

tested algorithms. This confirms the algorithm's strength in capturing nonlinear interactions while being less sensitive to noise and overfitting, as highlighted in ecological applications (Belgiu & Drăguţ, 2016; Räsänen et al., 2021; Cutler et al., 2007). GBM also showed strong performance, with low errors and consistent $R^2$ values, reflecting its capability to efficiently leverage key predictors (Kämäräinen et al., 2023; Natekin & Knoll, 2013). GAM, in contrast, had the weakest performance among all models at 1 m resolution, with the lowest $R^2$ (0.62), highest RMSE (0.077), and highest MAE (0.025). This likely reflects the model's limited ability to capture sharp spatial variability in $CH_4$ fluxes when localized structure is strong. GAMs rely on detecting smooth nonlinear effects, but when predictors become noisy or spatially complex, the fitted splines lack the detail needed for accurate prediction (Wood, 2017).

At 10 m resolution, RF not only achieved the lowest mean absolute and root-mean-square errors, but its $R^2$ and error metrics also changed the least when we varied resolution or added temporally dynamic predictors, indicating the most consistent performance in our experiments (Table 3).

GBM showed similarly low errors but a slightly lower $R^2$ (0.57). SVR achieved the highest $R^2$ (0.68), but this was offset by much higher prediction errors, indicating poor generalisation despite high apparent fit. GAM performed worst, with the lowest $R^2$ (0.53) and the highest RMSE (0.13).

Table 3. Performance of four models at 1 m and 10 m spatial resolutions. Metrics include $R^2$ (coefficient of determination), MAE (mean absolute error), and RMSE (root mean square error). Bold values represent the best score for each metric within each resolution. The "10 m" scenario includes models with temporally stable normalized difference vegetation index (NDVI) and normalized difference water index (NDWI), while "10 m_temporal" refers to models using temporally dynamic indices, matched to the closest available date of in-field $CH_4$ flux measurements.

| Model Type | Resolution | $R^2$ | MAE | RMSE |
|---|---|---|---|---|
| GAM | 1 m | 0.616 | 0.025 | 0.077 |
| | 10 m | 0.527 | 0.027 | 0.126 |
| | 10 m_temporal | 0.645 | **0.022** | 0.084 |
| GBM | 1 m | 0.625 | 0.008 | 0.012 |
| | 10 m | 0.570 | 0.008 | 0.013 |
| | 10 m_temporal | 0.689 | 0.117 | **0.024** |
| RF | 1 m | 0.744 | **0.006** | **0.010** |
| | 10 m | 0.650 | **0.007** | **0.012** |
| | 10 m_temporal | **0.751** | 0.016 | 0.105 |
| SVR | 1 m | **0.868** | 0.019 | 0.078 |
| | 10 m | **0.682** | 0.022 | 0.117 |
| | 10 m_temporal | 0.668 | **0.022** | 0.124 |

The decrease in SVR and GAM performance at 10 m resolution likely reflects the loss of fine-scale spatial detail when data are aggregated to coarser grids. At coarser resolution, each pixel represents a mixture of surface types and microtopographic conditions, which reduces local variability in the predictors and weakens the model's ability to capture small-scale relationships with $CH_4$ fluxes. SVR models, which depend on detailed nonlinear patterns, become less stable when this localised structure is smoothed out. Similarly, GAM performance declines when predictors become more homogeneous, since spline functions can no longer represent fine spatial gradients. In contrast, RF and GBM were more robust to this loss of detail because their ensemble structure allows them to generalise better under coarser input conditions. Based on these results, we selected RF and GBM for further analysis as the most reliable combination of accuracy and cross-resolution stability. When cross-validation was grouped by site or year, $R^2$ values dropped to ~0.1-0.2 and RMSE increased severalfold compared to standard CV, reflecting the lack of repeated measurements under identical environmental conditions across years (Text S4, Fig. S7).

Including the temporal variability of the NDVI and NDWI values led to an average increase in $R^2$ of approximately 0.11 for the GAM, GBM and RF models at a resolution of 10 m (Table 3). SVR was the exception, showing a slight decrease in $R^2$ with no reduction in errors. For the GAM model, this increase in explanatory power was accompanied by lower RMSE and MAE values, indicating more accurate and robust performance. In contrast, both RF and GBM showed a higher $R^2$, but also exhibited increased absolute errors, which may indicate overfitting to temporally dynamic predictors. This likely reflects the tendency of ensemble models to capture noise in dynamic inputs when training data is limited (Barry and Elith, 2006; Chollet Ramampiandra et al., 2023; Reichstein et al., 2019). Similar behaviour has been observed in other ecosystem carbon flux modelling studies, for example, in neural network models that overfit to lagged meteorological inputs (Papale & Valentini, 2003). The GAM model likely benefited from its ability to represent gradual ecological shifts through penalised smoothers, which reduces sensitivity to noise (Berbesi & Pritchard, 2023). The limited improvement in performance for SVR may be due to its sensitivity to data structure and lower flexibility when modelling smooth temporal trends in ecological datasets (Smola & Schölkopf, 2004).

### 3.3 Impact of model and resolution selection on CH$_4$ flux predictions

Because only meteorological variables changed over time, the interannual variation in the predicted maps arises from interactions between the static landscape predictors and varying atmospheric conditions, rather than from spatial changes in surface characteristics. Different data-driven models can produce distinct spatial predictions even when trained on the same input data. Although well documented, most machine-learning algorithms are not easily interpretable, whereas statistical approaches such as GAMs provide more transparent relationships between predictors and fluxes. We therefore compare their spatial predictions and simple diagnostics to assess reliability and guide model choice for CH$_4$ upscaling. Our comparison of upscaled CH$_4$ flux fields produced by the RF and GBM models showed that algorithm choice remained an important influence on spatial variability in predicted CH$_4$ fluxes (Fig. 6). The GBM model generated higher local contrast and more pronounced extremes, especially at 1 m resolution, with pronounced peaks in wet, topographically complex areas, reflecting its greater sensitivity to extreme values and local predictor variation. RF produced smoother, noise-resistant distributions, aligning with its known strength in generalising across heterogeneous landscapes (Räsänen et al., 2021; Cutler et al., 2007). While RF remains a robust and widely applied method for spatial upscaling (Cutler et al., 2007), our findings demonstrate that algorithm choice still affects spatial outcomes, with each model emphasising different aspects of landscape variability. This highlights the value of including multiple model types, not only for optimising performance, but also for quantifying model-driven uncertainty in CH$_4$ flux upscaling.

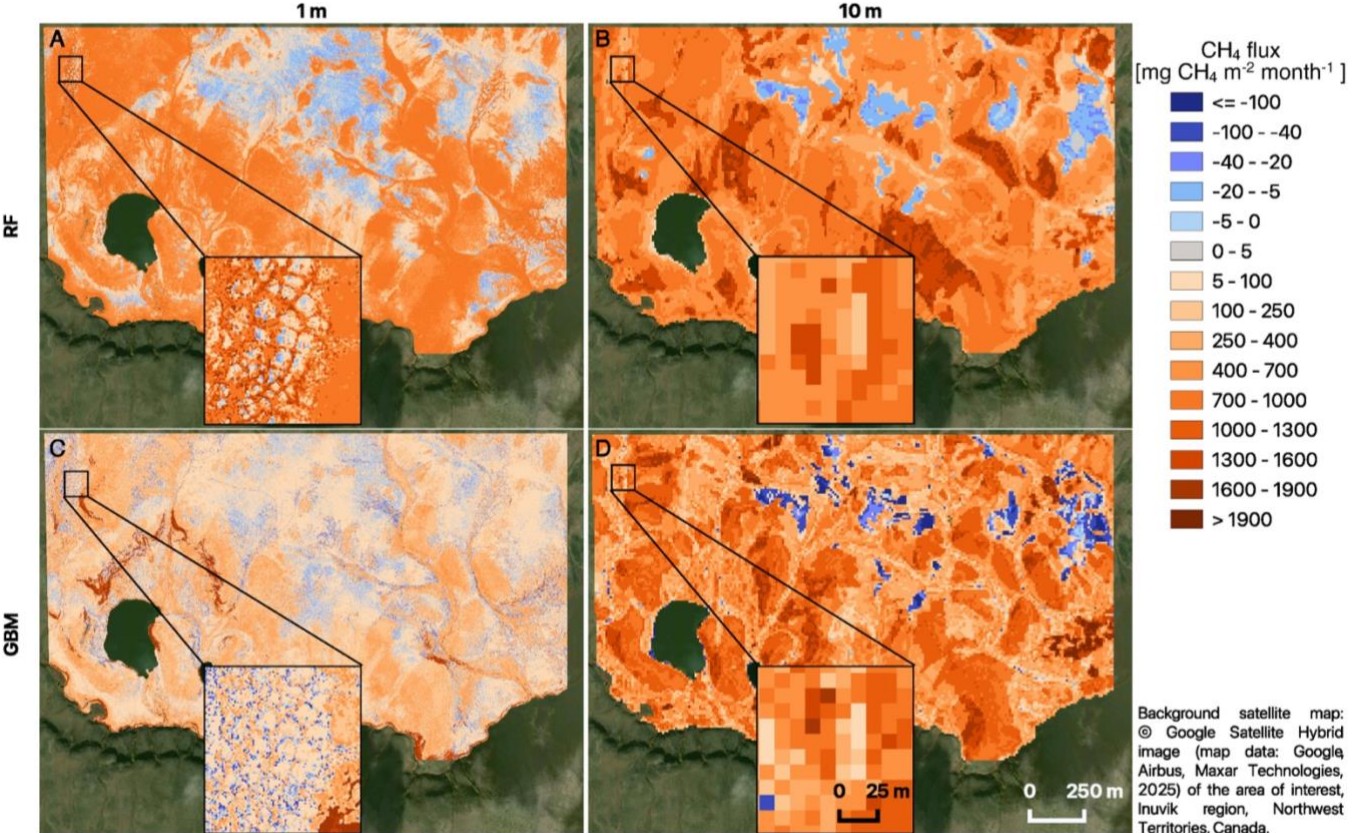

Figure 6. Predicted mean monthly CH₄ fluxes (mg CH₄ m⁻² month⁻¹) for July (averaged over 2019–2024), generated by two machine-learning models: Random Forest (RF, panels A and B) and Gradient Boosting Machine (GBM, panels C and D). The panels A and C shows predictions at 1 m resolution, and panels B and D right column at 10 m resolution. Each panel contains a black-framed zoom window, which enlarges a representative section of the polygonal mire. Visual comparison of the two insets illustrates how the fine wet-to-dry microtopography resolved at 1 m is smoothed when aggregated to 10 m. Background imagery: © Google Satellite Hybrid (Maxar, 2025).

Interestingly, although GBM exhibited more spatial flux variability, the mean fluxes predicted by GBM were consistently lower than those of RF. At 1 m, GBM averages 98.7 mg CH₄ m⁻² month⁻¹, whereas RF averages 518.6 mg; at 10 m the values rise to 608.8 mg and 683.4 mg, respectively (Fig. 7A). This more than fivefold difference at 1 m resolution underscores the substantial structural uncertainty that arises purely from algorithm selection, even when all predictors and training data are identical. At 10 m resolution, this discrepancy largely disappears because spatial aggregation smoothes microtopographic extremes and reduces the influence of local outliers, making both models converge toward similar mean fluxes. Net-sink pixels accounted for 10.0 % (RF) and 9.5 % (GBM) of the 1 m domain, but only 4.9 % (RF) and 4.4 % (GBM) at 10 m. CH₄ sink areas were spatially limited and highly sensitive to scale. Pixels acting as net CH₄ sinks (i.e. with negative monthly fluxes) were located on well-drained polygon rims and other lichen-dominated uplands where oxygen remained available throughout the summer. This allowed highly efficient methanotrophs to oxidise CH₄ faster than it was produced (Biasi et al., 2008). Resolving these units at a scale of 1 m showed that they covered around 10% of the scene and significantly reduced the landscape-mean flux. However, coarsening to 10 m mixed the aerobic patches with adjacent wet hollows, reducing their mapped extent to approximately 4.5% and erasing many uptake pixels. A comparable effect has been observed when chamber data were averaged across broader physiographic units, shifting site-level balances from weak sinks to slight sources (Zona et al., 2016). This pronounced scale effect is consistent with pan-Arctic syntheses, indicating that, although they cover only a small fraction of the surface, aerated uplands can offset a significant proportion of wetland emissions, yet they are often obscured in coarse products and regional budgets (Olefeldt et al., 2013; Kuhn et al., 2021). Our findings support recent assessments that retaining metre-scale information on microtopography, vegetation, and soil moisture is essential for capturing sink behaviour and ultimately for refining carbon budgets in permafrost regions, which currently indicate a small terrestrial CO₂ sink and a wetland CH₄ source (Treat et al., 2024).

Both models predict a comparable overall flux range, but the main disagreement occurs in the intensity and spatial extent of intermediate values. GBM also tends to produce stronger negative extremes, indicating higher sensitivity to localized sink conditions. The residual model disagreement is driven less by the number of sink pixels than by their intensity. Minimum fluxes predicted by GBM were consistently more negative than those from RF, with extremes of -147 mg $CH_4$ $m^{-2}$ month $^{-1}$ (1 m) and -330 mg $CH_4$ $m^{-2}$ month$^{-1}$ (10 m), compared to -45 and -33 mg $CH_4$ $m^{-2}$ month $^{-1}$ in RF, respectively. This suggests that GBM may emphasise $CH_4$ sink strength more than RF, even though the spatial extent of sinks is similar across models.

At 1 m, GBM often responds more strongly to localized environmental extremes. These include areas with much higher soil moisture, surface temperature spikes, or abrupt changed in microtopography that may only occur at the metre scale. This is due to its sequential learning process, which can emphasize subtle but high-impact predictors. RF, in contrast, smooths local extremes and yields more conservative area means. Because GBM-1 m produced a markedly lower AOI mean than RF, we treat this behaviour as a potential systematic bias toward stronger sinks and hotspots. We therefore use RF-1 m as the reference budget estimate and retain GBM-1 m as a sensitivity case to bracket structural uncertainty. At 10 m, aggregation reduces fine-scale contrasts and the RF-GBM predictions converge. Pixel-wise standard deviations (Fig. 7B) reveal that RF is temporally more stable, while GBM is more sensitive to inter-annual variation, particularly in wet or geomorphically complex areas.

Spatial differences between models and resolutions were calculated as pixel-wise subtraction (RF – GBM and 1 m – 10 m), ensuring consistent direction of comparison across all analyses. Additional analysis of spatial differences between models (Fig. B2) showed that several predictors were moderately to strongly correlated with the differences between RF and GBM predictions. At 1 m resolution, the strongest correlation was observed for NDWI (-0.53), indicating that model disagreement was most pronounced in wetter areas. NDVI (0.49) and landscape type (0.41) also showed strong positive correlations with model differences, suggesting greater divergence in vegetated zones and across cover transitions. For the 10 m products, aspect (0.43) became the only predictor for model differences above 0.4, implying that model choice matters most on directionally exposed terrain once fine micro-relief is lost. Across both resolutions, NDWI exhibited consistent negative correlations, implying that divergences are magnified in wetter and concave landforms that tend to accumulate water or thaw differently. These findings are in line with Tagesson et al. (2013), who showed that adding satellite-derived NDWI improves $CH_4$ flux modelling by capturing moisture-driven variability.

The full spatial difference maps for each model and resolution are provided in the supplementary Zotero dataset (Ivanova et al., 2025a) to enable direct comparison with environmental layers and visual exploration of model- and scale-driven patterns.

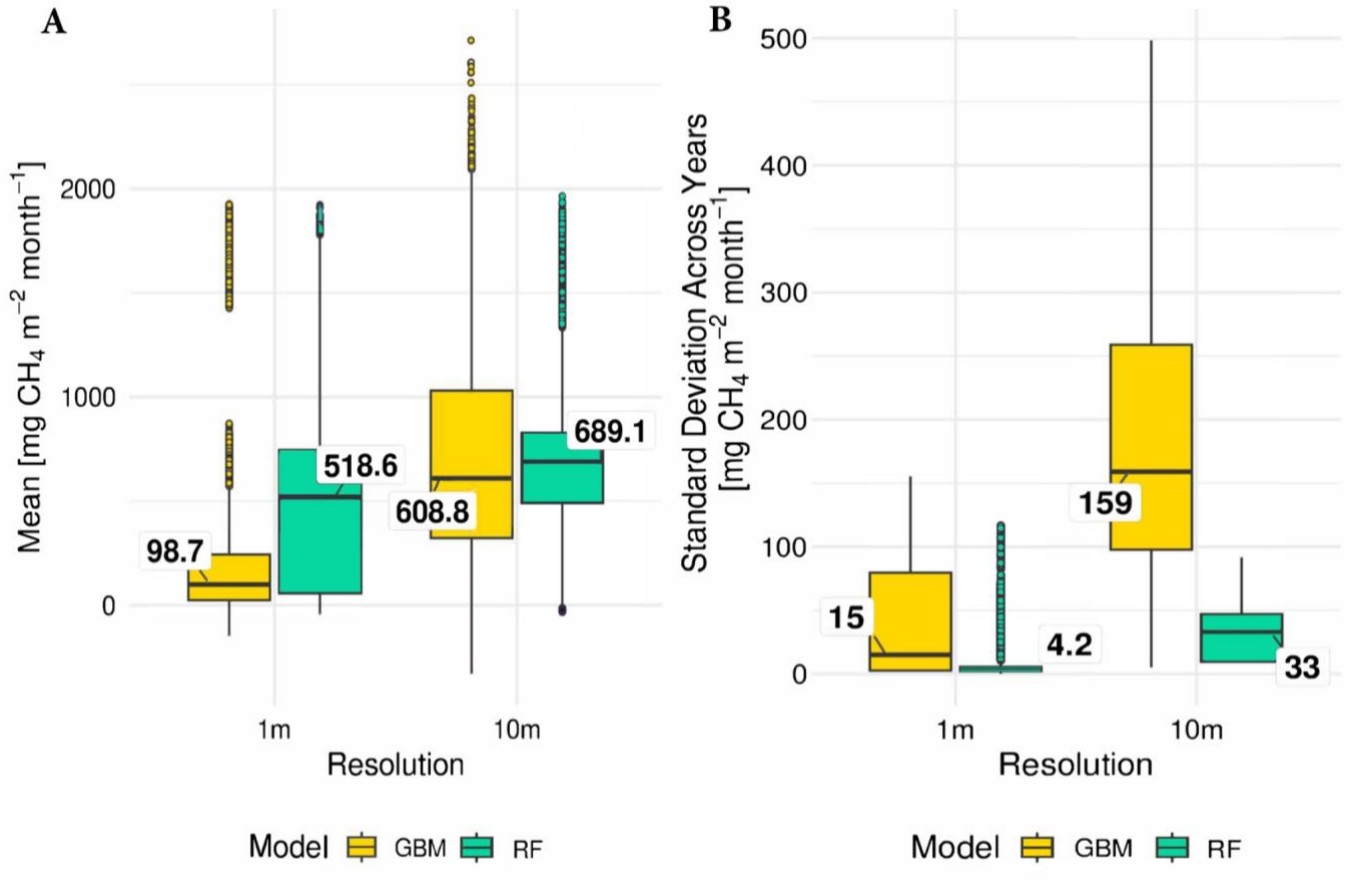

**Figure 7. Mean CH₄ flux and interannual variability across the study area. (A) Mean monthly CH₄ flux predicted by RF and GBM models, averaged over the entire area of interest for the period July 2019 to 2024. (B) Pixel-wise interannual standard deviation of predicted CH₄ fluxes for July months from 2019 to 2024, calculated separately for each model and resolution. Each boxplot shows the distribution of values across all pixels: the box spans the interquartile range (IQR, 25th to 75th percentile), the horizontal line within the box indicates the median, and whiskers extend to 1.5×IQR. Points beyond the whiskers represent potential outliers.**

Part of the disagreement between the two models, particularly at 10 m resolution, can be attributed to limited training data in certain landscape types such as tall-shrub and complex wetland zones, which were sampled less intensively due to access constraints. These classes show higher prediction uncertainty and stronger divergence between RF and GBM, as GBM amplifies local extremes while RF tends to smooth them.

Model performance based on the aggregated 1 m data (10 m from 1 m) was nearly identical to that of the original 10 m models, with only small differences across algorithms. GBM and SVR showed slightly improved accuracy after aggregation, while RF performed marginally worse and GAM remained nearly unchanged. These results indicate that the performance differences between the 1 m and 10 m models reported above are mainly attributable to spatial resolution rather than to differences between sensor-based and aggregated input data (see Text S4, Fig. S5, and Fig. S6).

### 3.4 Parameters importance in CH₄ flux prediction

Analysis of the relative importance of the predictors revealed fundamental differences between the RF and GBM models, and how these differences change when moving from 1 m to 10 m resolution (Table B3). Significance was assessed using the permutation method for each model and scale combination.

At the 1 m resolution, RF distributed importance fairly evenly across the topographic parameters. TPI (~22 %), Aspect (~21 %), and Slope (~18 %) showed comparably high influence, followed by landscape class (~16 %). All other predictors contributed less than 10 %, and meteorological drivers collectively stayed below that level. This topography-centred profile is consistent with the moderate intercorrelation among terrain metrics such as TPI, Slope, and TWI (Fig. B1), which share a common DEM origin and partly capture overlapping relief and moisture patterns. Such behaviour aligns with the known tendency of random

forests to distribute importance across correlated terrain drivers due to their random feature-selection mechanism (Räsänen et al., 2021; Cutler et al., 2007).

GBM showed a different pattern: again, no single parameter dominates, but five drivers spread across different input categories (Slope, Landscape class, AT, TDD and NDWI) each explained about 14-16 % of the total, and none exceeds 20 %. This flatter profile is based on the boosting process. Each new tree fixes the errors left by the previous one, so different predictors take turns
improving the model (Friedman, 2001). When several drivers reduce error by a similar amount, the model splits importance among them (Kämäräinen et al., 2023).

When the resolution was coarsened to 10 m, pixel aggregation smoothed micro-relief, and both algorithms shifted toward moisture-integrating drivers as primary explanatory influences. In RF, NDWI (~25 %) and Landscape class (~25 %) emerge as joint leaders, NDVI rises to ~12 %, and all topographic parameters drop below 8 %. In GBM, the re-organisation is even stronger:
the moisture indicators NDWI (~25 %) and TWI (~19 %) together explained almost half of the total importance, while landscape class follows at ~11 % and Slope and Aspect fall below 7 %. This pattern agrees with field evidence that moisture proxies dominate $CH_4$-flux prediction at coarser resolution, where fine-scale topographic details are lost (Tagesson et al., 2013; Wangari et al., 2023). NDWI and TWI both integrate water content over several pixels, making them potential surrogates for local water-table height and the extent of anoxic microsites that drive methanogenesis. NDWI is also sensitive to vegetation water and
phenology, allowing it to track water-table depth in peatlands (Meingast et al., 2014; Kalacska et al., 2018). TWI, which maps landscape-scale water accumulation and thus redox and gas-diffusion controls, aligns with syntheses showing that water-table fluctuations set the size of anoxic zones and largely govern $CH_4$ production and emission (Kaiser et al., 2018; Cui et al., 2024). Landscape class and NDVI contributed complementary information on vegetation type and biomass, which modulate both substrate supply and methane oxidation. In practical terms, upscaling to 10 m can still capture landscape-scale $CH_4$ patterns, but
only if robust moisture indices such as NDWI and TWI were included; purely geometric terrain drivers lose most of their explanatory power once microtopography is averaged out.

The potential influence of CALU, subsidence, and temporally matched NDVI/NDWI indices was further examined in a separate 10 m model experiment (Text S3, Fig. S3).

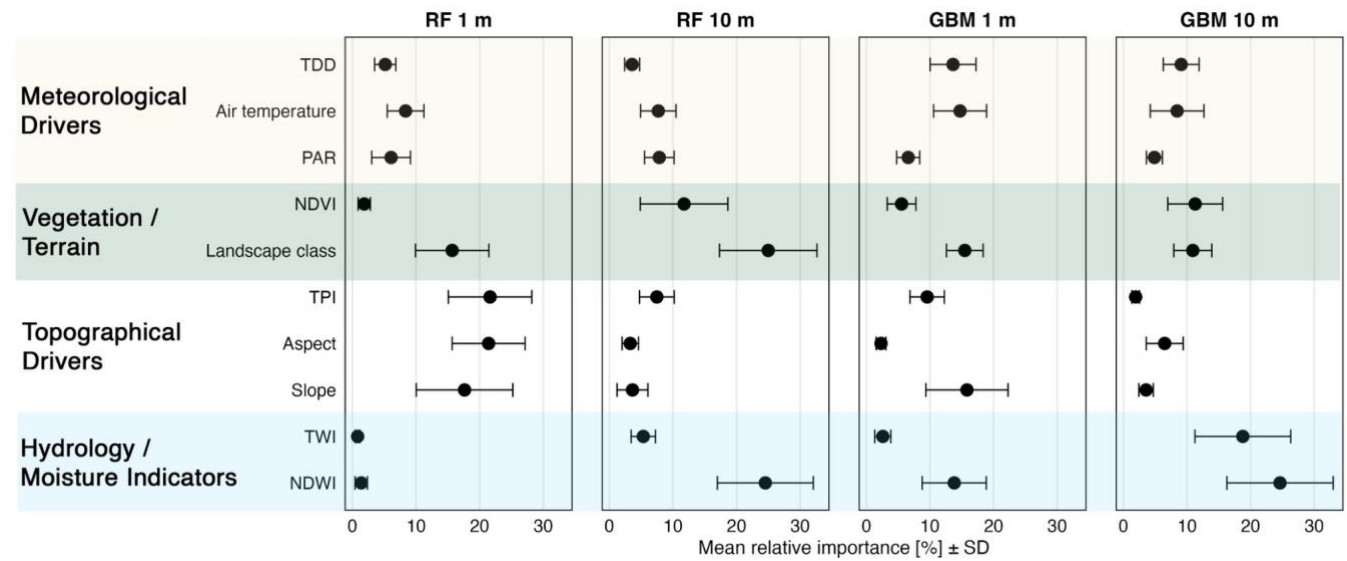

**Figure 8 Mean relative importance (± SD) of environmental predictors for CH₄ fluxes across two machine-learning models –Random Forest (RF) and Gradient Boosting Machine (GBM) – evaluated at 1 m and 10 m spatial resolutions. Importance was estimated by bootstrap resampling (n = 100) and is expressed as a percentage of total importance within each model. Predictors are grouped into four categories: Meteorological drivers (thawing degree days, air temperature, photosynthetically active radiation), Vegetation/Terrain (Normalized Difference Vegetation Index, landscape class), Topography (Topographic Position Index, aspect,
slope), and Hydrology/Moisture (Topographic Wetness Index, Normalized Difference Water Index). Abbreviations: TDD – thawing degree days; PAR – photosynthetically active radiation; NDVI – Normalized Difference Vegetation Index; TPI – Topographic Position Index; TWI – Topographic Wetness Index; NDWI – Normalized Difference Water Index.**

**4 Conclusion**

This study aimed to identify the key environmental and spectral drivers of $CH_4$ fluxes in heterogeneous Arctic tundra, evaluate how both model performance and predictor importance change with spatial resolution and across different data-driven models, and assess the implications for upscaling $CH_4$ fluxes.

Subsidence, derived from InSAR, showed the highest correlation with observed $CH_4$ fluxes of all the tested predictors, emphasising its value as a spatial proxy for soil moisture. It should therefore be included directly in $CH_4$ upscaling workflows, particularly in permafrost landscapes where moisture conditions were key drivers of fluxes.

Although different models varied significantly in their estimates, RF and GBM provided the most consistent and reliable upscaling results. At the highest spatial resolution, the two algorithms produced notably different flux magnitudes, reflecting structural uncertainty linked to how each model handles local extremes. However, their robustness should be verified through targeted sensitivity analyses, including tests with modified predictor sets, varied hyperparameters, and bootstrapped subsampling, to assess the stability of variable importance and model performance. Significance of model predictors was found to be strongly scale-dependent. At a resolution of 1 m, the models derived most of their explanatory power from microtopographic metrics, which capture the detailed elevation contrasts that distinguish between hummocks and hollows, as well as localising $CH_4$ hotspots. However, after aggregation to 10 m, these relief cues were diluted, causing a change in ranking: moisture proxies NDWI and TWI became the principal drivers, together accounting for almost half of the explained variance. This transition from terrain- to moisture-controlled importance highlights the fact that fine-scale mapping requires detailed topographic data, whereas regional upscaling must prioritize robust hydrological indices. For AOI budgets we report RF at 1 m resolution as the reference and use GBM at 1 m resolution as a sensitivity bound due to its amplification of metre-scale extremes. Spatial resolution emerged as the important factor determining the predictive power data-driven upscaled $CH_4$ flux patterns, exerting a stronger influence than model choice. At a resolution of 1 m, fine-scale heterogeneity was captured at a high degree of detail, making it possible for models to distinguish between local sources and sinks of $CH_4$. At 10 m, micro features merge into mixed pixels, boosting mean fluxes and variability. This resulted in fine-scale sinks and hotspots disappearing, and in some cases, fluxes being misclassified as a source of $CH_4$ in dry areas. Consequently, 10 m models produced higher mean fluxes and broader flux distributions. However, some of these high values may be due to mixed-pixel artefacts rather than true local emissions.

Our study findings imply that resolution is not simply a case of 'the higher, the better', and similarly, more complex ML methods may not necessarily yield better predictions. Although 1 m models captured fine-scale heterogeneity, 10 m models with temporally dynamic predictors improve explanatory power but increase prediction errors, likely due to overfitting to short-term fluctuations. This suggests that, in some cases, 10 m resolution models can outperform 1 m resolution ones, particularly when enhanced with well-timed spectral information – though caution is needed to balance fine-scale accuracy with broader spatial generalisability.

Although this study focuses on a single Arctic wetland complex at Trail Valley Creek, the workflow and findings are broadly transferable to other tundra environments. Ten-metre inputs from Sentinel-2 and ArcticDEM reproduce dominant moisture-control patterns typical of Arctic lowlands, while metre-scale (drone + LiDAR) layers reveal fine sink–source contrasts but require intensive data collection. Scale effects may vary across Arctic landscapes depending on topographic and vegetation complexity, and could differ in more homogeneous or highly dissected terrain. Because the models remain correlative and July-specific, extending the workflow across seasons and additional sites would strengthen generality and test the stability of the observed scale effects. Future work should expand sampling into underrepresented landscape and vegetation classes, high-emission zones, methane uptake regions, and winter fluxes, and incorporate temporally dynamic predictors. Integrating theory-guided time-series modelling approaches informed by ecological theory could enhance both the interpretability and accuracy of $CH_4$ forecasts under complex seasonal dynamics, particularly when data availability is limited.

## Appendix A. Predictors from remote sensing and meteorological data

**Table A1. Overview of predictor variables used in the CH4 flux models. This table lists all environmental predictor variables considered in the modelling framework. For each parameter, the spatial resolution (for remote sensing layers), source, short description, and formulas for calculations are presented (where applicable). Parameters are grouped into six thematic categories: Meteorological Drivers (e.g., PAR, AT, TDD), Vegetation / Land Cover (e.g., NDVI, landscape classification, CALU), Hydrology / Moisture Indicators (e.g., NDWI, TWI), Topography (e.g., slope, aspect, TPI), and Surface Deformation (subsidence). Each variable is marked as either static (unchanging during the study period) or dynamic (time-specific).**

| Parameter | spatial resolution | Derived from | Description | Temporal variability | Parameter type |
|---|---|---|---|---|---|
| Photosynthetically Active Radiation (PAR) | 1 km | NASA Langley Research Center (2024) | Extracted as a predictor variable for $CH_4$ flux models. | Dynamic | Meteorological Drivers |
| NDVI | 1 m | Rettelbach et al. (2024) | Ultra-high resolution NDVI derived from drone imagery. $NDVI = \frac{NIR - Red}{NIR + Red}$ (A1) | Static | Vegetation /Terrain |
| NDWI | 1 m | Rettelbach et al. (2024) | Ultra-high resolution NDWI derived from drone imagery. $NDWI = \frac{Green - NIR}{Green + NIR}$ (A2) | Static | Hydrology / Moisture Indicators |
| Landscape classification | 1 m | Rettelbach et al. (2024), Lange et al., 2021 | Landscape classification performed using 1 m drone imagery & ALS-derived DTM (Appendix B). | Static | Vegetation /Terrain |
| NDVI | 10 m | Sentinel-2 [2019 - 2024] (mean for July - August 2018). | Extracted from the composite Sentinel-2 image for July - August 2018. $NDVI = \frac{NIR - Red}{NIR + Red}$ (A1) | Static | Vegetation /Terrain |
| NDWI | 10 m | Sentinel-2 [2019 - 2024] (mean for July - August 2018). | Extracted from the composite Sentinel-2 image for July - August 2018. $NDWI = \frac{Green - NIR}{Green + NIR}$ (A2) | Static | Hydrology / Moisture Indicators |
| NDVI | 10 m | Sentinel-2 [2019 - 2024] (Single-date, closest to flux measurement). | Extracted from single-date, closest to flux measurement. $NDVI = \frac{NIR - Red}{NIR + Red}$ (A1) | Dynamic | Vegetation /Terrain |
| NDWI | 10 m | Sentinel-2 [2019 - 2024] (Single-date, closest to flux measurement). | Extracted from single-date, closest to flux measurement. $NDWI = \frac{Green - NIR}{Green + NIR}$ (A2) | Dynamic | Hydrology / Moisture Indicators |
| Landscape classification | 10 m | Copernicus Sentinel-2 data [2018], ArcticDEM v4 (Porter et al., 2023) | Landscape classification performed using Sentinel-2 indices (2018) and terrain derivatives of ArcticDEM (Appendix B). | Static | Vegetation /Terrain |
| Slope | 1 m | Lange et al., 2021 | Measures the rate of elevation change along the steepest descent. It controls water and material flow, influences soil moisture, erosion, and formation, and is a | Static | Topographical parameters |

| Parameter | spatial resolution | Derived from | Description | Temporal variability | Parameter type |
|---|---|---|---|---|---|
| | | | key hydrological and geomorphological factor. Derived from DTM. | | |
| Aspect | 1 m | Lange et al., 2021 | Represents the compass direction a slope faces, measured in degrees clockwise from north. It influences microclimate, solar radiation, snowmelt, and vegetation patterns. Derived from DTM. | Static | Topographical parameters |
| TWI | 1 m | Lange et al., 2021 | TWI combines upslope catchment area and slope to model potential soil moisture accumulation. It is commonly used to identify areas potentially prone to saturation and water accumulation. $TWI = ln\frac{a}{tan\ b}$, - a = *upslope contributing area per unit contour length* - b = *local slope angle* Derived from DTM. | Static | Topographical parameters |
| TPI_30m | 1 m | Lange et al., 2021 | The Topographic Position Index (TPI) quantifies the elevation of a cell relative to the mean elevation of surrounding cells, allowing differentiation between ridges, valleys, and flat areas. We computed TPI using a 30 m circular moving window, meaning that for each location, its elevation was compared to the average of all surrounding elevations within a 30 m radius. This window size smooths out small-scale variation and captures broader landform patterns. Derived from DTM. | Static | Topographical parameters |
| Slope | 10 m (from 2 m) | ArcticDEM v4 (Porter et al., 2023) | Measures the rate of elevation change along the steepest descent. It controls water and material flow, influences soil moisture, erosion, and formation, and is a key hydrological and geomorphological factor. Derived from DTM. | Static | Topographical parameters |
| Aspect | 10 m (from 2 m) | ArcticDEM v4 (Porter et al., 2023) | Represents the compass direction a slope faces, measured in degrees clockwise from north. It influences microclimate, solar radiation, snowmelt, and vegetation patterns. Derived from DTM. | Static | Topographical parameters |
| TWI | 10 m (from 2 m) | ArcticDEM v4 (Porter et al., 2023) | TWI combines upslope catchment area and slope to model potential soil moisture accumulation. It is commonly used to identify areas potentially prone to saturation and water accumulation. | Static | Topographical parameters |

| Parameter | spatial resolution | Derived from | Description | Temporal variability | Parameter type |
|---|---|---|---|---|---|
| | | | $TWI = ln\frac{a}{tan\,b}$, (A3) <br> - a = *upslope contributing area per unit contour length* <br> - b = *local slope angle* <br><br> Derived from DTM. | | |
| TPI_30m | 10 m (from 2 m) | ArcticDEM v4 (Porter et al., 2023) | The Topographic Position Index (TPI) quantifies the elevation of a cell relative to the mean elevation of surrounding cells, allowing differentiation between ridges, valleys, and flat areas. We computed TPI using a 30 m circular moving window, meaning that for each location, its elevation was compared to the average of all surrounding elevations within a 30 m radius. This window size smooths out small-scale variation and captures broader landform patterns. Derived from DTM. | Static | Topographical parameters |
| Subsidence | 10 m | Copernicus Sentinel-1/2 data | Seasonal deformation has been derived from Sentinel-1 time series (2018 - 2023) using SAR Interferometry. Six years have been averaged to reduce noise. The seasonal deformation rates in thawing degree days domain represent near surface soil moisture spatial patterns. (Widhalm et al., 2025) | Static | Surface Deformation |
| CALU | 10 m | CALU <br><br> (Bartsch et al., 2024) | The Circumarctic Landcover Units provide a consistent high-resolution land cover classification across the entire Arctic tundra. CALU defines **23 units of similar reflectance derived from multispectral (Sentinel-2) and C-band SAR (Sentinel-1) data**. The classification reflects wetness gradients, shrub density, moss abundance, and surface moisture (Bartsch et al., 2024). | Static | Vegetation /Terrain |
| AT | Point | Trail Valley Creek meteorological station (Climate ID: 220N005; WMO ID: 71683; TC ID: XTV). | Hourly air temperature measured at 2 m above ground level. Used as a dynamic meteorological driver for $CH_4$ flux models. | Dynamic | Meteorological Drivers |
| Thawing Degree Days (TDD) | Point | Trail Valley Creek meteorological station (Climate ID: 220N005; | Cumulative positive air temperature sum (above 0 °C) used as a proxy for thaw energy and season length. Calculated per flux measurement period | Dynamic | Meteorological Drivers |

| Parameter | spatial resolution | Derived from | Description | Temporal variability | Parameter type |
|---|---|---|---|---|---|
| | | WMO ID: 71683; TC ID: XTV). | based on air temperature from meteorological station. $$TDD = \sum_{i=1}^{n} max(T_{mean,i}, 0) \ ,$$ (A4) <br> <ul><li>$T_{mean,i}$ = mean daily air temperature on day $i$</li><li>n = number of days in the accumulation period</li><li>The **max** function ensures only temperatures above 0 °C are counted</li></ul> | | |

**Text A1. Landscape classification**

To classify land cover in the TVC area, we employed a supervised classification approach using multi-source remote sensing data at 1 m and 10 m resolutions. The classification process was implemented in Google Earth Engine (GEE), enabling large-scale data processing. A Random Forest (RF) classifier was chosen due to its ability to handle high-dimensional data, its resistance to overfitting, and its suitability for land cover mapping. By applying a consistent classification framework at both 1 m and 10 m resolutions, this study enables direct comparisons of classification performance across spatial scales,

Training and Validation Data

The classification was trained using manually collected validation points that were assigned to six distinct land cover classes: Dwarf Shrub, Tall Shrub, Sedges, Tussock, Lichen, and Water. To ensure statistical robustness, 80 % of the validation points were used for model training, while the remaining 20 % were reserved for accuracy assessment.

Remote Sensing Data and Feature Extraction

To optimise classification accuracy, we integrated spectral, texture, and topographic features derived from multiple remote sensing sources. Sentinel-2 optical imagery at 10 m resolution was used for broad-scale classification, with images acquired during the 2018 growing season (25 June - 4 September 2018) to ensure that differences in land cover classification were due to spatial resolution rather than changing environmental conditions, matching the same summer period as the 1 m drone survey. Topographic features were extracted from ArcticDEM (2 m resolution) (Porter et al., 2023). At finer spatial scales, we incorporated ultra-high resolution drone imagery (1 m and 10 cm) from Rettelbach et al. (2024) and a digital terrain model (DTM) (Lange et al., 2021).

To further enhance classification accuracy, we performed a Gray-Level Co-occurrence Matrix (GLCM) texture analysis of NDVI, allowing us to incorporate information on vegetation heterogeneity. A 2 × 2 kernel was used for 10 m classification, while a 20 × 20 kernel was applied at 1 m resolution to capture 20 m spatial patterns.

**Table A2. Parameters used for the landscape classification. Abbreviations in the table: NDVI – Normalized Difference Vegetation Index, NDWI – Normalized Difference Water Index, EVI – Enhanced Vegetation Index, SAVI – Soil-Adjusted Vegetation Index, GLCM – Gray-Level Co-occurrence Matrix, TPI – Topographic Position Index, TWI – Topographic Wetness Index, DEM – Digital Elevation Model. Spectral indices were derived from Sentinel-2 (10 m spatial resolution) and drone imagery (1 m spatial resolution) using the visible and near-infrared bands (Blue, Green, Red, NIR).**

| Parameter | Description | Formula (if applicable) | Spatial resolution |
|---|---|---|---|
| NDVI | Measures vegetation greenness | $\dfrac{NIR - RED}{NIR + RED}$ | 10 m, 1 m |
| NDWI | Identifies water and moisture content | $\dfrac{Green - NIR}{Green + NIR}$ | 10 m, 1 m |
| EVI | Improves sensitivity to high biomass | $2.5 \times \dfrac{NIR - RED}{NIR + 6 \times RED - 7.5 \times Blue + 1}$ | 10 m, 1 m |
| SAVI | Reduces soil brightness effects | $\dfrac{(NIR - RED) \times (1 + L)}{NIR + RED + L}$, where $L = 0.5$ | 10 m, 1 m |
| GLCM Entropy | Measures randomness in pixel intensity | Derived from NDVI | 10 m, 1 m |
| GLCM Contrast | Captures local texture variation | Derived from NDVI | 10 m, 1 m |
| GLCM Homogeneity | Measures uniformity in image texture | Derived from NDVI | 10 m, 1 m |
| Slope | Measures terrain steepness | Derived from DEM | 2 m, 1 m |
| Aspect | Identifies terrain orientation | Derived from DEM | 2 m, 1 m |

| Parameter | Description | Formula (if applicable) | Spatial resolution |
|---|---|---|---|
| TPI 6m | Detects local terrain position | Elevation - Mean(Elevation within 6m radius) | 2 m, 1 m |
| TPI 30m | Identifies broader-scale landforms | Elevation - Mean(Elevation within 30m radius) | 2 m, 1 m |
| TWI | Estimates soil moisture potential | $ln(\frac{A}{tan(\beta)})$, where<br>A = specific contributing area<br>$\beta$ = slope in radians | 2 m, 1 m |
| Band parameters | Captures spectral variation in different wavelengths | mean and sd for each pixel of RGB and NIR bands | 10 m |
| Band parameters | Captures spectral variation in different wavelengths | pixel value of RGB and NIR bands | 10 m |

Classification Model and Accuracy Assessment

The Random Forest classifier was trained separately for 10 m Sentinel-2 data and 1 m drone-based data, with 200 decision trees

used in both cases. The trained models were then applied to classify the entire dataset. The overall accuracy was 0.76 for 1 m

resolution and 0.71 for 10 m resolution. Class-specific accuracies are provided in Table S1.

Export

Final classified maps at 10 m and 1 m resolutions were exported as GeoTIFF files for further analysis and comparison.

**Table A3. Description of Circumarctic Land Cover Units (CALU) present in the study area. Class names and definitions are taken**
**from Bartsch et al. (2024). Additional columns indicate (i) whether the class is present within the area of interest (AOI), and (ii) whether**
**CH$_4$ flux measurements are available for this class.**

| CALU class | Description | Present in AOI | CH$_4$ measurements available |
|---|---|---|---|
| 1 | Water | yes | |
| 2 | shallow water/abundant macrophytes | yes | |
| 3 | wetland, permanent | yes | yes |
| 4 | wet to aquatic tundra (seasonal), abundant moss | yes | |
| 5 | moist to wet tundra, abundant moss, prostrate shrubs | | |
| 6 | dry to moist tundra, partially barren, prostrate shrubs | yes | |
| 7 | dry tundra, abundant lichen, prostrate shrubs | | |
| 8 | dry to aquatic tundra, dwarf shrubs (& sparse tree cover along treeline) | yes | |
| 9 | dry to moist tundra, prostrate to low shrubs | yes | yes |
| 10 | moist tundra, abundant moss, prostrate to low shrubs | yes | yes |
| 11 | moist tundra, abundant moss, dwarf, and low shrubs | yes | yes |
| 12 | moist tundra, dense dwarf, and low shrubs (& sparse tree cover along treeline) | yes | |
| 13 | moist to wet tundra, dense dwarf, and low shrubs (& sparse tree cover along treeline) | yes | |
| 14 | moist tundra, low shrubs | yes | |
| 15 | dry to moist tundra, partially barren | yes | yes |
| 16 | moist tundra, abundant forbs, dwarf to tall shrubs | yes | |
| 17 | recently burned or flooded, partially barren | yes | |
| 18 | forest (deciduous) with dwarf to tall shrubs | yes | |

| 19 | forest (mixed) with dwarf to tall shrubs | yes | |
|----|------------------------------------------|-----|---|
| 20 | forest (needle leave) with dwarf and low shrubs | yes | |
| 21 | partially barren | yes | |
| 22 | snow/ice | | |
| 23 | other (incl. shadow) | yes | |

**Appendix B. Results**

**Table B1. Summary statistics of observed CH₄ fluxes (mg CH₄ m⁻² h⁻¹) across site-specific landscape classes at 1 m and 10 m spatial resolutions. The table reports the number of observations (N Obs), number of sites, where measurements were done (N sites), mean, first quartile (Q1), third quartile (Q3), minimum, and maximum CH₄ flux values for each landscape class at both resolutions.**

| Landscape class | Resolution | N Obs | N sites | Mean | Q1 | Q3 | Min | Max |
|---|---|---|---|---|---|---|---|---|
| Lichen | 1 m | 1713 | 22 | 0.002 | -0.02 | 0 | -0.48 | 1.62 |
| | 10 m | 3690 | 19 | -0.011 | -0.02 | 0 | -0.48 | 0.62 |
| Tussock | 1 m | 11372 | 39 | -0.016 | -0.03 | -0.01 | -0.24 | 2.41 |
| | 10 m | 9218 | 30 | -0.020 | -0.03 | -0.01 | -0.18 | 0.68 |
| Dwarf shrub | 1 m | 130 | 4 | 0.053 | -0.02 | 0.06 | -0.18 | 0.89 |
| | 10 m | 201 | 7 | 0.19 | -0.01 | 0.13 | -0.28 | 2.41 |
| Tall shrub | 10 m | 71 | 3 | 0.024 | -0.02 | 0.05 | -0.12 | 0.54 |
| Sedges | 1 m | 177 | 3 | 0.94 | 0.06 | 1.09 | -0.02 | 6.39 |
| | 10 m | 204 | 9 | 0.87 | 0.05 | 1.07 | -0.03 | 6.39 |

**Table B2. Summary statistics of observed CH₄ fluxes (mg CH₄ m⁻² h⁻¹) across CALU classes. The table reports the number of observations (n), mean, first quartile (Q1), third quartile (Q3), minimum, and maximum CH₄ flux values for each landscape class at both resolutions. Class descriptions are available in Bartsch et al. (2024).**

| CALU class | n | Mean | Q1 | Q3 | Min | Max |
|---|---|---|---|---|---|---|
| 3. Permanent wetland | 11 | -0.09 | -0.09 | 0 | -0.48 | 0 |
| 9. Dry to moist tundra, prostrate to low shrubs, tussocks | 6357 | -0.01 | -0.02 | 0 | -0.19 | 0.61 |
| 10. Moist tundra, abundant moss, prostrate to low shrubs, tussocks | 6407 | -0.02 | -0.03 | -0.01 | -0.18 | 0.41 |
| 11. Moist tundra, abundant moss, dwarf and low shrubs, tussocks | 490 | 0.46 | 0.01 | 0.39 | -0.28 | 6,39 |
| 15. Moist to wet tundra, abundant lichen, in some cases partially barren (disturbed). | 119 | 0 | 0 | 0.03 | -0.24 | 0.06 |

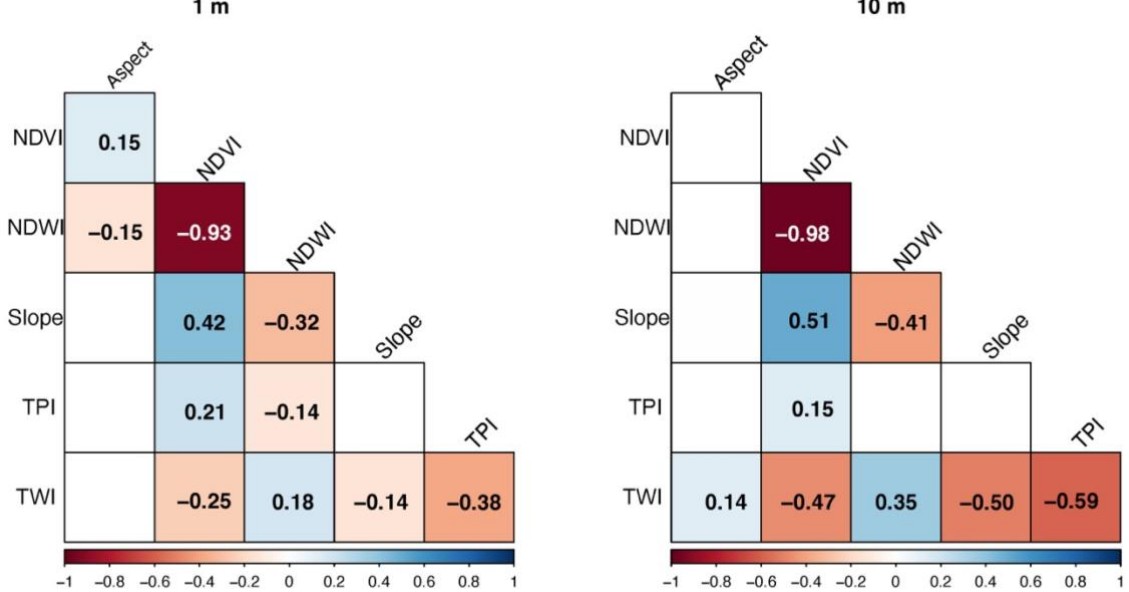

**Figure B1. Spearman rank correlations between environmental predictors used in the CH₄ flux models at (left) 1 m and (right) 10 m resolution. Only statistically significant relationships (p < 0.05) with absolute correlation strength |ρ| > 0.1 are shown; non-significant and weak correlations are blanked. Positive correlations are shown in blue and negative correlations in red, with colour intensity proportional to correlation strength (see scale bar, −1 to +1)**

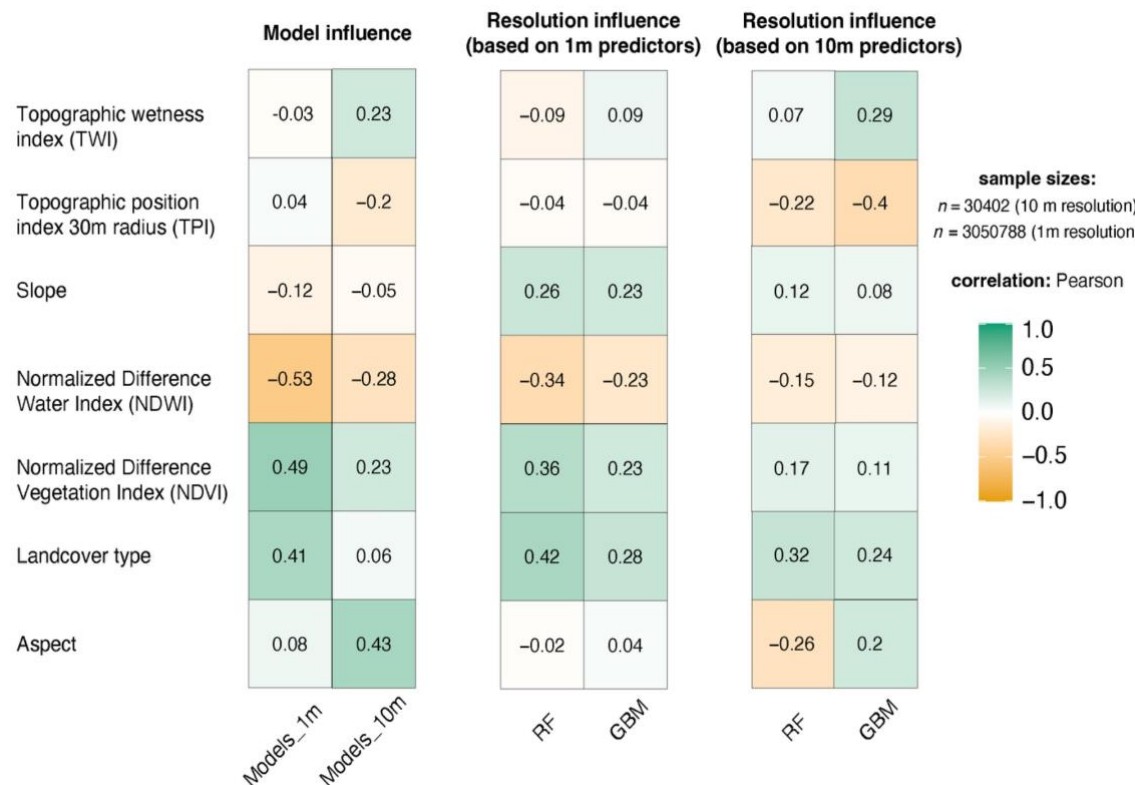

**Figure B2. Pearson correlation between spatial differences in CH₄ flux predictions and selected environmental predictors. "Model influence" (left block) shows differences between RF and GBM predictions at the same resolution (RF − GBM). "Resolution influence" (middle and right blocks) show differences between 1 m and 10 m predictions (1 m − 10 m), calculated using predictors derived from (i) the 10 m products downscaled to 1 m (middle) and (ii) the 1 m predictors aggregated to 10 m (right). Positive correlations indicate that higher predictor values coincide with stronger CH₄ flux mismatches between models or resolutions. Each cell represents Pearson's r across 30 402 pixels (10 m) and 3 050 788 pixels (1 m).**

**Table B3. Relative importance [%] of environmental predictors for CH$_4$ flux models across spatial resolutions and algorithms. The table shows the variable importance (in %) for each predictor derived from Random Forest (RF) and Gradient Boosted Machine (GBM) models at 1 m and 10 m spatial resolution. Predictors are grouped by thematic category (e.g., Meteorological, Topographic). Importance values reflect the mean contribution of each predictor to the model performance and standard deviations (± SD).**

| Group | Parameter | RF 1 m | RF 10 m | GBM 1 m | GBM 10 m |
|---|---|---|---|---|---|
| **Meteorological Drivers** | Air temperature | 8.4 ± 2.9 | 7.7 ± 2.8 | 14.7 ± 4.2 | 8.4 ± 4.2 |
| | PAR | 6.1 ± 3 | 7.8 ± 2.3 | 6.6 ± 1.8 | 4.9 ± 1.3 |
| | TDD | 5.2 ± 1.7 | 3.6 ± 1.2 | 13.6 ± 3.6 | 9.1 ± 2.8 |
| **Hydrology / Moisture Indicators** | NDWI | 1.4 ± 1 | 24.5 ± 7.5 | 13.8 ± 5 | 24.6 ± 8.4 |
| | TWI | 0.8 ± 0.3 | 5.3 ± 1.9 | 2.6 ± 1.3 | 18.8 ± 7.5 |
| **Topographical parameters** | Aspect | 21.4 ± 5.7 | 3.3 ± 1.3 | 2.3 ± 0.8 | 6.5 ± 2.9 |
| | Slope | 17.6 ± 7.6 | 3.6 ± 2.4 | 15.8 ± 6.5 | 3.6 ± 1.1 |
| | TPI | 21.6 ± 6.6 | 7.5 ± 2.7 | 9.6 ± 2.7 | 1.9 ± 0.6 |
| **Vegetation / Terrain** | NDVI | 1.9 ± 1 | 11.7 ± 6.9 | 5.5 ± 2.3 | 11.3 ± 4.3 |
| | Landscape class | 15.7 ± 5.8 | 25 ± 7.7 | 15.5 ± 2.9 | 10.9 ± 3 |

**Author contributions**

KI, MG, AB, BW, VB, and TS received funding from the European Research Council Synergy Grant Q-Arctic (grant no. 951288). MG further acknowledges financial support from the EU HORIZON EUROPE programme (project GreenFeedBack, grant agreement number 101056921), the European Space Agency (AMPAC-net project, contract number 4000137912/22/I-DT) and the German Ministry for Research, Technology and Space (MOMENT project, Grant No. 03F0931G). AV received support from the Gordon and Betty Moore Foundation (grant no. 8414), and funding catalyzed by the TED Audacious Project (Permafrost Pathways). CV received financial support from the Research Council of Finland project MUFFIN (no. 332196) and the European Research Council Starting Grant COLDSPOT (no. 101163177). OS acknowledges financial support through the Canada Research Chair (CRC-2018-00259) and NSERC Discovery Grants program (DGPIN-2018-05743 awarded, ArcticNet, a Network of Centres of Excellence Canada (grant no. P216), Canada First Research Excellence Fund's Global Water Futures program (Northern Water Futures), and the Polar Continental Shelf Program (608-20 and 602-21).

**Competing interests**

The authors declare that they have no conflict of interest.

**Acknowledgments**

We are grateful for the support of all research teams and the assistance provided at Trail Valley Creek Research Station in organizing the fieldwork. We are grateful to Dr. Theresia Yazbeck for insightful feedback that helped improve the manuscript. Parts of the text were language-edited with DeepL, and some analysis and plotting scripts were revised for clarity with assistance from ChatGPT (OpenAI; accessed January 2025). All outputs were reviewed and verified by the authors, who take full responsibility for the content.

We acknowledge the Trail Valley Creek orthophotos, digital surface models, and 3D point clouds provided by the MOSES airborne campaign and published by Rettelbach et al. (2024) on PANGAEA (DOI: 10.1594/PANGAEA.961942). ArcticDEM provided by the Polar Geospatial Center under NSF-OPP awards 1043681, 1559691, 1542736, 1810976, and 2129685. We acknowledge the use of Copernicus Sentinel-2 data (Contains modified Copernicus Sentinel data [2018 - 2024]). We acknowledge the use of NASA POWER data, provided by the NASA Langley Research Center (LaRC) through the POWER Project, funded by the NASA Earth Science/Applied Science Program.

**Financial support**

KI, MG, AB, BW, VB, and TS received funding from the European Research Council Synergy Grant Q-Arctic (grant no. 951288). AV received support from the Gordon and Betty Moore Foundation (grant no. 8414), and funding catalyzed by the TED Audacious Project (Permafrost Pathways). CV received financial support from the Research Council of Finland project MUFFIN (no. 332196) and the European Research Council Starting Grant COLDSPOT (no. 101163177). OS acknowledges financial support through the Canada Research Chair (CRC-2018-00259) and NSERC Discovery Grants program (DGPIN-2018-05743 awarded, ArcticNet, a Network of Centres of Excellence Canada (grant no. P216), Canada First Research Excellence Fund's Global Water Futures program (Northern Water Futures), and the Polar Continental Shelf Program (608-20 and 602-21). Further support was provided by the European Space Agency through the AMPAC-Net project (contract no. 4000137912/22/I-DT).

**Code availability**

The code used for $CH_4$ flux modelling, resolution comparison, and upscaling across Arctic wetland landscapes is publicly available on Zenodo (Ivanova et al., 2025b): https://doi.org/10.5281/zenodo.15399084

**Data availability**

The $CH_4$ flux predictions, spatial difference maps, and input dataset used in this study are publicly available on Zenodo (Ivanova et al., 2025a): https://doi.org/10.5281/zenodo.15753253

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
