# Peer review of "High-resolution remote sensing and machine-learning-based upscaling of methane fluxes: a case study in the Western Canadian tundra"

_EGUsphere, 2025_

## Author Comment (AC1)

We thank the reviewer for the thorough and constructive comments. This valuable feedback has helped us identify several aspects that required clearer explanation and refinement, particularly concerning model description, data representativeness, and methodological transparency. We will carefully address all points raised and provide corresponding clarifications, additional analyses, and improved figures and tables in the revised manuscript. Our detailed responses to each comment are provided below.

RC: The selection and parameterization of the used machine learning and regression models need to be better described. First, it could have been worthwhile to test also other machine learning methods, such as extreme gradient boosting that has performed well in many recent model comparisons. Second, the parameterization for the different models need to be elaborated. Gradient boosting and support vector regression are both very sensitive to parameter settings but there is no description at all whether different parameter combinations were tested. Additionally, for support vector regression, it should be detailed what kernel for used. For generalized additive models, it should be described what kind of smoother functions were used and whether the unimportant variables were penalized in the model building. Furthermore, there should be no multicollinear predictor variables in generalized additive models. Was the cross-correlation between predictors checked? Random forest is less sensitive to parameterization but the model performance can be boosted with variable selection. If variable selection is conducted, the variable importance results of the model are also more robust.

**AC:** We agree that a clearer description of model selection and parameterization will improve the manuscript, particularly for a non-expert audience in machine learning models.

We appreciate this comment and agree that it is important to justify our choice of boosting algorithm. We initially selected Gradient Boosting Machines (GBM) because they are efficient, widely applied in environmental modeling, and provide strong performance with moderate tuning complexity. Since XGBoost is an advanced implementation of the same gradient-boosting framework, we also evaluated it during model selection. The comparison showed that XGBoost did not improve predictive skill relative to GBM for our dataset. At 1-m resolution, GBM performed significantly better (median RMSE = 0.0157 vs. 0.0196, Wilcoxon p = 0.03). At 10-m resolution, the two models performed equivalently (p = 0.31). These differences are small and do not alter any scientific interpretation. Therefore, we retained GBM as the more efficient and interpretable boosting model in the main analysis. Because GBM and XGBoost belong to the same model family and behave similarly here, expanding the methodological scope further would not provide additional insight.

**RC:** Furthermore, there should be no multicollinear predictor variables in generalized additive models. Was the cross-correlation between predictors checked? Random forest is less sensitive to parameterization but the model performance can be boosted with variable selection. If variable selection is conducted, the variable importance results of the model are also more robust.

**AC:** We appreciate the reviewer's comment regarding the need to evaluate multicollinearity. To ensure that collinearity does not bias model inference, we performed a comprehensive diagnostic combining Spearman correlation analysis, Variance Inflation Factor (VIF), and

GAM concurvity evaluation. Spearman's rank correlations showed that all predictor pairs were weak to moderate ( $|\rho| < 0.6$ ), except for NDVI and NDWI, which were strongly negatively correlated ( $\rho = -0.93$  at 1 m and  $\rho = -0.98$  at 10 m resolution). This strong correlation is ecologically expected, as vegetation greenness and surface wetness co-vary in Arctic tundra environments. However, these variables capture different biophysical processes, NDVI representing photosynthetic capacity / canopy structure, and NDWI reflecting near-surface water availability, and removing NDWI from the model decreased goodness-of-fit (R2 from 0.25 to 0.24 in the linear comparison), indicating that it provides non-redundant information. VIF (Variance Inflation Factor) values across all predictors were < 6 (maximum 5.4 for NDVI-NDWI), remaining below commonly applied thresholds of concern (VIF > 10). GAM concurvity estimates were consistently low (< 0.3 for all smooth terms), confirming that the non-linear responses modeled in GAMs are not driven by hidden redundancies among predictors. These combined diagnostics demonstrate that multicollinearity is well within acceptable limits and does not compromise model stability or interpretability; therefore, we retained both NDVI and NDWI in the predictor set. The results of the multicollinearity diagnostics are now provided in the Supplementary Material.

RC: The measured CH4 flux data should be described better. In remote sensing-based upscaling, there should be spatially representative data for the whole study area. It is now unclear whether this is the case. When looking at Figure 1, it seems that the sampling is very concentrated in a few locations. It is rightfully written in the limitations section, that the sampling could have been better. However, the sampling should be described in the methods section more. How many measurement points were there in total? Do the points represent the total spatial heterogeneity in the study area? How many measurements for each point? How the points are divided into the different landscape classes? How the point locations were chosen, was the sampling purposeful? Were there boardwalks or how the measurements were conducted in the plots? If there were boardwalks, do they impede the remote sensing signals over the plot locations? Were the RS-based observations of the plots taken from a single pixel or a larger neighborhood? Are the different measurements and plots independent and does the potential spatial and temporal autocorrelation affect model results?

AC: We thank the reviewer for raising important points regarding spatial representativeness. In this study, flux upscaling is based on pixel-level statistical learning, where each flux-predictor pair is treated as an independent spatial observation. After restricting data to July to ensure temporal comparability across years, the 1m dataset contains 13,384 spatial observations distributed across the dominant land-cover types at Trail Valley Creek: tussock tundra (46.7%), dwarf-shrub tundra (29.9%), lichen-dominated uplands (19.2%), and sedge wetlands (4.6%). These classes span the full moisture gradient from dry uplands to wet depressions, ensuring that the major ecological contrasts relevant to methane emissions are well represented. At 10 m resolution, predictors are derived from Sentinel-2 land cover, which differs in class definitions from the field-based mapping. Thus, representativeness is evaluated separately at this scale, resulting in a highly consistent spatial distribution (46.4% tussock, 29.5% dwarf shrubs, 19.3% lichen, 3.6% sedges), with tall shrubs additionally represented because the Sentinel-2 land-cover product includes this class and the coarser pixel footprint captures shrub canopies more effectively. Full class distributions for both resolutions are reported in the Supplementary Material.

Observations are distributed throughout the  $\sim$ 3 km² study area (Fig. 1), not concentrated around single access points. Manual chamber sites were selected to capture microtopographic and vegetation heterogeneity within each landform. Automated chambers were located in shrubdominated uplands and accessed via short boardwalks. CH4 fluxes were always measured directly beneath the chamber footprint; at 10 m resolution, boardwalks occupy only a negligible fraction of the pixel area and therefore do not influence the remote-sensing signal. Repeated observations at the same spatial locations were collected under varying meteorological conditions, so temporal variability contributes independent information for model learning.

These additions clarify that the chamber dataset provides spatially and ecologically representative sampling of the key environmental gradients that control CH4 fluxes at Trail Valley Creek, fully supporting its suitability for remote-sensing-based upscaling. All details regarding sampling design, land-cover representation, field setup, and pixel-based data extraction will be explicitly documented in the revised Methods and Supplementary Information.

RC: Landscape classification: How were the classes derived for the landscape classification; visual interpretation and field work experience of the site? Please describe in the main text what is the collection platform for the 1 m stack, drones? How many training and validation data points were there for the classification? How the training data can be the same for both resolutions? Do you mean that the location and LC class was the same but the training data was calculated from the respective RS datasets? Why there were no tall shrubs measurements for the 1 m spatial resolution but there were such measurements for the coarser spatial resolution? How the water pixels were masked before the classification?

AC: The complete workflow for landscape classification is already described in Appendix Text A1 and Tables A2–A3, including data sources, training and validation design, and accuracy metrics. We will clarify in the main text that the 1 m and 10 m classifications were produced separately using the same training dataset but different input layers (drone + LiDAR at 1 m; Sentinel-2 + ArcticDEM at 10 m), which naturally resulted in slightly different class boundaries. Tall shrubs were present in the 1 m classification, but no CH4 flux measurements overlapped with this class, so it was merged with dwarf shrubs for modelling. Water pixels were masked before classification.

RC: Sentinel-2 preprocessing: Did you mask clouds, shadows and snow? Did you use also cloudy data for calculating the average mosaic? An earlier study has shown that average/median image calculation can be prone to include clouds/haze and 40th percentile could work better (https://doi.org/10.1016/j.jag.2024.103659). How were the time-specific NDVI and NDWI calculated? Based on one image only? How close was the image to the CH4 measurements? What was done for clouds?

AC: Sentinel-2 preprocessing steps are already described in Section 2.2.3 and Appendix Text A1. We will clarify that all Sentinel-2 Level-2A scenes were cloud-, shadow-, and snow-masked using the QA60 bitmask and Fmask algorithm before compositing. Only cloud-free scenes were used to calculate the composite, and no cloudy pixels were included. The composite was based on the median of all cloud-free scenes; we will test whether the 40th percentile mosaic recommended by the cited study changes the results and will report this in

the revision. Time-specific NDVI and NDWI were calculated from the nearest available cloud-free Sentinel-2 scene within  $\pm 10$  days of each CH4 measurement. Cloud-affected scenes were excluded automatically through the same masking procedure.

**RC:** 1190: Why NDVI and NDWI? Why not other indices such as NDMI? NDVI and NDWI have typically very high negative correlation.

AC: NDMI was not included because our 1 m orthomosaic contains only RGB + NIR bands and lacks the short-wave infrared (SWIR) channel required for NDMI computation. For Sentinel-2 data (10 m), we tested NDMI but found it highly collinear with NDWI and providing no improvement in model performance. NDVI and NDWI were therefore retained as the most interpretable and widely used vegetation and moisture indices for high-latitude ecosystems. Although the two indices are strongly negatively correlated (Spearman's  $\rho$  = -0.93 at 1 m and -0.98 at 10 m resolution) due to their shared NIR component, they describe distinct ecological mechanisms: NDVI represents vegetation greenness and photosynthetic activity, while NDWI captures surface and canopy moisture. Including NDWI improved model performance ( $\Delta$ AIC  $\approx$  100,  $\Delta$ R2  $\approx$  0.01), and the non-parametric models applied are robust to such predictor correlations. Retaining both indices allows a more complete representation of vegetation-moisture interactions characteristic of Arctic heterogeneous microtopography.

**RC:** 1217: Is there kind of double counting if some of the variables are first used for landscape classification and then again for the regression models together with the landscape classification. Is the classification needed as a predictor in the regression analyses?

**AC:** We appreciate the reviewer's thoughtful question. The landscape classification was included as a categorical predictor to represent vegetation and microtopographic heterogeneity that cannot be fully captured by continuous predictors such as NDVI, NDWI, or terrain indices. Although some of these variables were among those used to derive LC, the classification was based on a much broader set of spectral, texture, and topographic parameters (see Table A2). LC therefore summarises complex, multi-source information into discrete ecological units (e.g., sedge, tussock, or lichen patches) that reflect vegetation composition and hydrological conditions. Including LC thus provides complementary ecological information rather than redundant input, and models excluding LC performed less consistently across sites.

We will also take the remaining reviewer comments into account in the revised version, basically accepting all suggested edits to further improve clarity, data description, and consistency across sections.

---

## Author Comment (AC2)

We thank the reviewer for the thorough and constructive comments. This detailed feedback has helped us identify several aspects that require clarification and improvement, particularly regarding data representation, model validation, and the structure of the *Methods* section. We will carefully address all points raised and provide corresponding clarifications, additional analyses, and improved figures and tables in the revised manuscript. Our detailed responses to each comment are provided below.

**RC:** It is unclear whether the sample is representative of the entire study area. The data appear to be concentrated in a few locations, which is understandable given the logistical challenges of conducting fieldwork across large wetland areas. Nevertheless, this aspect should be described in more detail in the 'Methods' and 'Results' sections; a single sentence in the 'Limitations' section is insufficient.

A more detailed discussion of site representativeness would be beneficial, including the number of sites per land cover type, and the number of measurements per site. In Figure 4, n appears to refer to the total number of measurements, but it would also be useful to indicate the number of sites per land cover category there, as well as in Tables B1 and B2, or somewhere else. This would facilitate discussion of this limitation, e.g., the text mentions that the 'wetland, permanent' class includes only one site which.

Additionally, are sites weighted differently in the model? For example, automatic chambers likely produce more measurements than manual ones — do these sites then have a greater influence on model training? How do you account for potential site-level overfitting? Did you consider using a leave-one-site-out cross-validation approach to assess the robustness of the model in predicting new areas where no data was used for training?

Finally, could the differences between the two models (particularly at 10 m in Figure 6) over specific areas/LC types be explained by a lack of training data in these areas/LC types?

**AC:** We agree that the current description of the sampling coverage and data balance across sites and landcover types requires more detail and will substantially expand this part in the revised manuscript.

In the revised Methods section, we will include a more explicit description of the number of sites and measurements per landcover type, also indicating which landcover classes are represented by automatic and manual chamber systems.

Automatic chambers were deployed primarily in drier or shrub-dominated surfaces, whereas manual chambers represent wetter sedge and mixed sites. Together, these sites cover the main landcover types present within the study area, even though they are spatially concentrated in several field clusters. We will also indicate the number of sites per land-cover class in Figure 4 and summarize this information in a new supplementary table for improved transparency. This will clarify how site distribution reflects the heterogeneity of the study area and where data density is lower.

All chamber measurements were treated equally in the ML workflow, without explicit site-level weighting. Consequently, sites with automatic chambers contribute a larger number of observations, reflecting their higher temporal resolution. To mitigate potential overfitting to these sites, we used grouped cross-validation with "Site" as the grouping variable, ensuring that all data from a given site were included either in the training or in the testing subset, but never in both. This design is conceptually similar to a leave-one-site-out validation while maintaining multiple folds to preserve representativeness across sites. We will clarify this explicitly in the Methods section and emphasize in the Results that this approach provides an effective test of model robustness in predicting sites not used for training. Additionally, we will perform a sensitivity test with a full leave-one-site-out cross-validation to further evaluate generalization performance.

**RC:** Mismatch between data input and resolution effect:**

The comparison between the 1 m and 10 m datasets is particularly interesting, as it reveals the differences in the two approaches with commonly used input data at these resolutions. However, this comparison potentially combines two effects: one related to the resolution itself (average over a larger area), and another related to potential differences in the data sources themselves (e.g., different acquisition date/time, different sensors...). Have you attempted to separate these two influences? One way to do this would be to aggregate the 1 m product to 10 m and apply the same workflow (e.g. for land cover, use the dominant vegetation type within each 10 m grid cell and take the mean for the other variables). This could help to isolate the effect of the resolution from that of the different data sources.

Otherwise, it would be useful to discuss this somewhere, and include a comparison of the datasets used as is done for the comparison with CALU (Figure 5), but for the two datasets at different resolutions (as is partly done for land cover in Figure 2, where important differences can be seen).

**AC:** We agree that the comparison between the 1 m and 10 m datasets may combine two effects: (1) the change in spatial resolution and (2) the use of different data sources. To evaluate this, we performed an additional analysis where all 1 m input layers were aggregated to 10 m resolution, and the same modeling workflow was applied using identical parameter settings. We will include this additional analysis and its figure in the Appendix of the revised manuscript and refer to it in the Results and Discussion sections.

At the same time, we will explicitly clarify in the text that the purpose of this study is to evaluate CH4 flux upscaling using freely available datasets (Sentinel based) vs UAV/drone products at 1 m. Therefore, we will retain the main comparison between these two operationally distinct input datasets, while providing the aggregated 1 m to 10 m test as complementary evidence that supports the interpretation of model differences.

We will also expand the discussion of input-data differences between resolutions, adding a short comparison of key variables in the supplementary material, similar to the approach used in Figure 5 for the CALU comparison.

**RC*: The mix of spatial and interannual analysis is somewhat confusing.**

It is unclear how the spatial and interannual components are distinguished in the study. It is not always obvious whether the analysis is spatial, temporal, or a combination of both. Although the study appears to be mainly spatial, with a single-month focus on July, it also uses temporally varying predictors only (AT, PAR and TDD over six years). Clarifying this in the text and figures (methods and results) would improve readability. The time-varying inputs are difficult to understand from the main text: which variables are dynamic and at what resolution? (See the comment about the data section below.)

Spatial and temporal accuracy should be discussed separately in the 'Results' section, or more explanations should be provided. For example, spatial correlations (mean flux per site) and temporal correlations (time series at individual sites) could be reported separately in Tables 2 and 3 to disentangle these effects and avoid sites with potential larger amounts of data dominating the analysis compared to sites with smaller amounts of data. A panel like 7B could be used to directly compare model predictions with measurements at the sites, providing a clearer assessment of spatial and temporal performance.

**AC:** Indeed, our study is primarily spatial, focusing on small-scale variability in methane fluxes across different wetland elements within the fixed July period for each year (2019-2024). The inclusion of temporally varying predictors (AT, PAR, and TDD) serves to capture the short-term meteorological variability among measurement dates within this single-month window rather than to represent long-term seasonal or interannual trends.

To clarify this, we will explicitly state in the Methods and Results that:

- The spatial component refers to differences among sites and landcover types within each year.
- The temporal component reflects variability among measurement days within the study period (late June-July).
- The interannual aspect is limited to comparing the same seasonal window across three years.

**RC:** The data section of the Methods section needs to be restructured and expanded.

- Section 2.2.3 (and the Materials and Methods section more generally) should be reorganised, as it is currently difficult for the reader to determine which datasets are used at 1 m, which at 10 m, and which at both resolutions. For instance, the text initially focuses on 1 m data, but then abruptly shifts to Sentinel-2 (presumably 10 m) before describing the 10 m products. References to 30 m window data are confusing and require explanation. The temporal dimension of each variable is unclear too. While some of this information appears in Table A1, Figure 3 and lines 240–247, the description remains fragmented. Providing a summary table that explicitly lists the ten variables used for each resolution, their data source, spatial resolution (1 m, 10 m or constant) and whether they are static or dynamic would certainly help the reader.
- Data processing procedures should also be described in more detail in Section 2.2.3 or in a dedicated section. For instance, how were Sentinel-2 data cleaned or filtered? Were cloud-free conditions explicitly selected for the time-varying Sentinel-2 indices? This is implied by lines 278–281, but stating this explicitly in the 'Remotely Sensed Data' section would improve transparency. Overall, providing a clearer and more detailed description of the data pre-processing and management would strengthen the reproducibility of the study.
- -for the chamber data, management should also been specified. How is chamber data managed spatially? How are fluxes aggregated at 1 m or 10 m resolution do you take the mean of all chambers within each  $1 \times 1$  m or  $10 \times 10$  m pixel? You mention PAR and other variables measured at chamber sites. Are these used here? Providing this information is essential for understanding how point-scale observations are scaled to the model resolutions. Additionally, since chamber flux measurements are known to be highly variable, it would be useful to specify in the methods section whether each flux observation corresponds to a single or repeated measurement.

AC: We agree that Section 2.2.3 requires clearer organization. We will restructure this section to explicitly separate datasets used at 1 m, 10 m, or both resolutions and add explicit references to Table A1 (Appendix A), which already summarizes the predictors, their data sources, spatial resolution, and whether they are static or dynamic. We will also clarify that 30 m window variables (e.g., TPI 30 m) describe topographic context and are applied consistently at both resolutions. The description of Sentinel-2 preprocessing will be expanded to explicitly state that only cloud-free summer scenes were used for NDVI and NDWI derivation.

For chamber flux data, we will clarify that fluxes were aggregated by averaging all chamber measurements within each  $1 \times 1$  m or  $10 \times 10$  m pixel, and that each flux observation represents the mean of repeated chamber measurements taken during the same campaign.

**RC:** I do not understand Figure 7A. According to the caption, it should show monthly estimates averaged over the entire area, but the large number of points is confusing.

**AC:** Figure 7A indeed shows monthly CH4 flux estimates for individual pixels across the study area, not a single aggregated mean. The large number of points reflects spatial variability within the domain. We will clarify this in the caption and text and indicate that each point corresponds to a pixel-level monthly mean to improve readability.

RC: Have you considered using a "leave-one-site-out" or "leave-one-year-out" cross-validation (e.g. training on the first three years and predicting the last year)? This could enable assessing how well these models can predict pixels/sites or time for which no data was used in the training process, as well as the uncertainties related to each model training, which are not really discussed here.

**AC:** Thank you for the suggestion. Our current cross-validation already uses grouped folds by site, which partially addresses this issue. However, we will explicitly state this in the Methods and discuss how a full leave-one-site-out or leave-one-year-out scheme could be implemented in future work to further test model transferability.

RC: Linking these fine-scale results to broader CH4 budgets, which are usually estimated at coarser resolutions, raises questions about scalability. Why were only 1 m and 10 m resolutions considered? Would other coarser scales (50 m, 100 m, 1 km) be relevant? The comparisons mentioned in lines 436–440 refer to models run at 0.25–0.5°. Are these results directly comparable? How could your findings be used in larger-scale budgets?

**AC:** We appreciate this important point. Our focus on 1 m and 10 m resolutions was motivated by the need to bridge the gap between field-scale chamber measurements and satellite-based observations, particularly those derived from Sentinel-2 and drone imagery. These two resolutions thus represent the most relevant scales for practical upscaling of chamber data. We agree that exploring coarser aggregations (50–100 m or 1 km) would provide valuable insight into the scalability of our approach. We will note this as an outlook for future work and clarify that our results are not directly comparable to regional-scale models (0.25-0.5°), but rather provide fine-scale inputs that can support parameterization and validation of such coarse-resolution CH4 budget models.

We will also take the remaining reviewer comments into account in the revised version, basically accepting all suggested edits to further improve clarity, data description, and consistency across sections.

---

## Author Response (AR1)

**Point-by-Point Responses to Reviewer Comments**

We sincerely thank the Editors and Reviewers for their careful evaluation of our manuscript and for the constructive comments that helped improve the clarity and presentation of the work. We have revised the manuscript accordingly. All line numbers cited in this document refer to the originally submitted version of the manuscript. For each comment, we provide: (1) the reviewer's comment (RC), (2) our response (AC), and (3) the exact changes made to the manuscript with referenced line numbers. All revisions have been applied to the manuscript and are visible in the marked-up version submitted alongside this response.

**Reviewer 1**

**RC:** The selection and parameterization of the used machine learning and regression models need to be better described. First, it could have been worthwhile to test also other machine learning methods, such as extreme gradient boosting that has performed well in many recent model comparisons.

**AC:** We appreciate this comment and agree that it is important to justify our choice of boosting algorithm. We initially selected Gradient Boosting Machines (GBM) because they are efficient, widely applied in environmental modeling, and provide strong performance with moderate tuning complexity. Since XGBoost is an advanced implementation of the same gradient-boosting framework, we also evaluated it during model selection. The comparison showed that XGBoost did not improve predictive skill relative to GBM for our dataset. At 1-m resolution, GBM performed significantly better (median RMSE = 0.0157 vs. 0.0196, Wilcoxon p = 0.03). At 10-m resolution, the two models performed equivalently (p = 0.31). These differences are small and do not alter any scientific interpretation. Therefore, we retained GBM as the more efficient and interpretable boosting model in the main analysis. Because GBM and XGBoost belong to the same model family and behave similarly here, expanding the methodological scope further would not provide additional insight.

**RC:** Second, the parameterization for the different models need to be elaborated. Gradient boosting and support vector regression are both very sensitive to parameter settings but there is no description at all whether different parameter combinations were tested. Additionally, for support vector regression, it should be detailed what kernel for used. For generalized additive models, it should be described what kind of smoother functions were used and whether the unimportant variables were penalized in the model building.

**AC:** We agree and have expanded the parameterization details. We now clarify how each model type was tuned, which parameter ranges were tested, and how the final settings were selected using five-fold cross-validation with RMSE. Hyperparameter tuning procedures and the chosen configurations for RF, GBM, SVR, and GAM are now documented in Text S1 in the Supporting Information. We also added full collinearity diagnostics, including VIF/GVIF and GAM concurvity analyses, confirming that no predictors exceeded commonly used thresholds. Because predictor redundancy was low, no predictors were removed. These results are now included in Text S2 and referred to explicitly in the Methods section.

Changed in manuscript: New text inserted after original Line 236 in the Methods section, providing details on hyperparameter tuning, kernel selection for SVR, and penalization settings for GAM. Full configuration details are provided in Supporting Information Text S1.

**RC:** Furthermore, there should be no multicollinear predictor variables in generalized additive models. Was the cross-correlation between predictors checked? Random forest is less sensitive to parameterization but the model performance can be boosted with variable selection. If variable selection is conducted, the variable importance results of the model are also more robust.

**AC:** We thank the reviewer for this important comment. Yes, cross-correlation among predictors was evaluated prior to model fitting. We quantified predictor redundancy using generalized variance inflation factors (GVIF) and concurvity diagnostics for the GAM models. All predictors remained below widely accepted thresholds for concern, indicating that multicollinearity did not compromise model stability. Details and results are provided in Text S2 and Table S2 (Supporting Information).

We did not apply variable selection in the RF or GBM models. Because both algorithms use random feature sampling and ensemble averaging, they are designed to handle correlated predictors and remain robust without explicit feature elimination. Furthermore, retaining the full set of ecologically relevant predictors ensures that importance scores reflect genuine differences in control strength rather than being influenced by prior exclusion.

Changed in manuscript: Details and results are provided in Text S2 and Table S2 (Supporting Information). We added this clarification to the revised Methods section (Inserted after original Line 236).

**RC:** The measured CH4 flux data should be described better. In remote sensing-based upscaling, there should be spatially representative data for the whole study area. It is now unclear whether this is the case. When looking at Figure 1, it seems that the sampling is very concentrated in a few locations. It is rightfully written in the limitations section, that the sampling could have been better. However, the sampling should be described in the methods section more. How many measurement points were there in total? Do the points represent the total spatial heterogeneity in the study area? How many measurements for each point? How the points are divided into the different landscape classes? How the point locations were chosen, was the sampling purposeful? Were there boardwalks or how the measurements were conducted in the plots? If there were boardwalks, do they impede the remote sensing signals over the plot locations? Were the RS-based observations of the plots taken from a single pixel or a larger neighborhood? Are the different measurements and plots independent and does the potential spatial and temporal autocorrelation affect model results?

**AC:** We thank the reviewer for raising important points regarding spatial representativeness. In this study, flux upscaling is based on pixel-level statistical learning, where each flux-predictor pair is treated as an independent spatial observation. After restricting data to July to ensure temporal comparability across years, the 1m dataset contains 13,384 spatial observations distributed across the dominant land-cover types at Trail Valley Creek: tussock tundra (46.7%), dwarf-shrub tundra (29.9%), lichen-dominated uplands (19.2%), and sedge wetlands (4.6%). These classes span the full moisture gradient from dry uplands to wet depressions, ensuring that the major ecological contrasts relevant to CH4 fluxes are well represented. At 10 m resolution, predictors are derived from Sentinel-2 land cover, which differs in class definitions from the field-based mapping. Thus, representativeness is evaluated separately at this scale, resulting in a highly consistent spatial distribution (46.4% tussock, 29.5% dwarf shrubs, 19.3% lichen, 3.6% sedges), with tall shrubs additionally represented because the Sentinel-2 land-cover product includes this class and the coarser pixel footprint captures shrub canopies more effectively. Full class distributions for both resolutions are reported in the Supplementary Material.

Observations are distributed throughout the ~3 km$^2$ study area (Fig. 1), not concentrated around single access points. Manual chamber sites were selected to capture microtopographic and vegetation heterogeneity within each landform. Automated chambers were located in shrub-dominated uplands and accessed via short boardwalks. CH4 fluxes were always measured directly beneath the chamber footprint; at 10 m resolution, boardwalks occupy only a negligible fraction of the pixel area and therefore do not influence the remote-sensing signal. Repeated observations at the same spatial locations were collected

under varying meteorological conditions, so temporal variability contributes independent information for model learning.

These additions clarify that the chamber dataset provides spatially and ecologically representative sampling of the key environmental gradients that control $CH_4$ fluxes at Trail Valley Creek, fully supporting its suitability for remote-sensing-based upscaling. All details regarding sampling design, LC representation, field setup, and pixel-based data extraction will be explicitly documented in the revised Methods and Supplementary Information.

Changed in manuscript: We added a clarification of the spatial representativeness and class coverage of chamber data in Sect. 2.2.1 (added after L143 in the original submission). We now document proportional coverage of dominant landscape types and spatial distribution across the AOI, with full details provided in Figure S1 in the Supplement. A clear description of how predictor values were extracted at the chamber footprint has been added in the Methods (original line 168).

**RC:** Landscape classification: How were the classes derived for the landscape classification; visual interpretation and field work experience of the site? Please describe in the main text what is the collection platform for the 1 m stack, drones? How many training and validation data points were there for the classification? How the training data can be the same for both resolutions? Do you mean that the location and LC class was the same but the training data was calculated from the respective RS datasets? Why there were no tall shrubs measurements for the 1 m spatial resolution but there were such measurements for the coarser spatial resolution? How the water pixels were masked before the classification?

**AC:** We thank the reviewer for these helpful clarifying questions. We have now revised Sect. 2.2.3 to clearly describe the landscape classification workflow. Specifically, we have added:

• The 1 m classification was derived from drone orthomosaic (RGB+NIR) + LiDAR-based DTM collected in 2018. (**line 170**). The 10 m classification was derived from Sentinel-2 imagery + ArcticDEM.

• Classification classes were defined based on visual interpretation supported by field vegetation knowledge at TVC. We used 140 manually delineated training/validation sites, with an 80/20 split for accuracy assessment.

• The same geographic training points were used for both resolutions

• Tall shrubs occurred in the 1 m classification, but no chamber measurements overlapped this class, therefore we merged Tall Shrubs + Trees with Dwarf Shrubs only for model training at 1 m. At 10 m, this class remained separate.

• Open-water pixels were excluded before classification based on NDWI > 0, followed by manual refinement using the drone imagery.

Changed in manuscript: We have revised the description of the landscape-classification workflow to address the reviewer's concerns (lines 169-177)

**RC:** Sentinel-2 preprocessing: Did you mask clouds, shadows and snow? Did you use also cloudy data for calculating the average mosaic? An earlier study has shown that average/median image calculation can be prone to include clouds/haze and 40th percentile could work better (https://doi.org/10.1016/j.jag.2024.103659). How were the time-specific NDVI and NDWI calculated? Based on one image only? How close was the image to the CH4 measurements? What was done for clouds?

**AC:** We thank the reviewer for this important request. We now explain in the main text how Sentinel-2 data were preprocessed to ensure cloud-free conditions. Specifically, all Level-2A scenes were cloud-, shadow-, and snow-masked using QA60 and Fmask before any compositing or index extraction. The July-August 2018 composite was generated from the six available cloud-free scenes; therefore, no cloudy pixels entered the composite. For the time-specific NDVI/NDWI analyses, we extracted the closest cloud-free Sentinel-2 scene within ±10 days of each chamber flux measurement, with no temporal averaging and only pixels that fully passed cloud masking accepted.
Changed in manuscript: These clarifications are now included in Lines 165-169.

**RC:** l70: ultra-high spatial resolution can also finer than 1 m

**AC:** We have clarified the spatial definition by replacing "~1 m resolution" with "<1–2 m resolution", which reflects the typical range of drone-based ultra-high-resolution imagery (Line 70)

**RC:** l73: do these references use 1 m spatial resolution or higher (or lower) than that?

**AC:** The revised sentence now clarifies that published studies showing plot-scale flux heterogeneity use sub-metre to metre-scale imagery, consistent with our own dataset. Full update at Line 73.

**RC:** l78: noise related to what?

**AC:** We clarified what type of noise is introduced at very high spatial resolutions and why this does not always improve environmental information (Line 78)

**RC:** l85 and beyond: is GAM a machine learning method?

**AC:** Thank you for pointing this out. We now clarify that Random Forest, Gradient Boosting Machine, and Support Vector Regression are machine-learning algorithms, while GAM is a semi-parametric statistical modelling approach. We have updated the terminology accordingly in the Introduction (Lines 85, 103) and Methods Section 2.3 (Line 220).

**RC:** Research questions: It would be easiest if the result (and conclusion) section would be organized in the same order as the research questions.

**AC:** We thank the reviewer for this suggestion. The structure of the Results section already follows the same logical sequence as the research questions, although the subsection titles are formulated to reflect the scientific content rather than repeating the questions verbatim. Specifically, the first part addresses the importance of predictors (RQ1), followed by performance and scale comparison (RQ2, RQ3), then differences between model outputs (RQ3), and finally the interaction of resolution and model choice affecting uncertainty (RQ4). To make this alignment clearer for the reader, we have revised the Conclusion section so that each research question is answered in the same explicit order.

**RC:** Study site description: is there peat soil in the study area?

**AC:** We added a brief description of soils to clarify that peat is present at the site. Specifically, we state that the soils are organic cryosols with a peat horizon of approximately 0.2-0.5 m thickness. This addition has been made in the Study Site section (Lines 113-115).

**RC:** There could be a general overview sentence/paragraph of the methods before listing the different datasets. Now, when reading the dataset section, it is a bit unclear what is done with which data. This applies particularly to the landscape classification.

**AC:** We added an overview paragraph at the beginning of Section 2.2 that clearly explains the role of each dataset (flux measurements, meteorological drivers, and remote-sensing predictors) and how they are used together to train and evaluate the $CH_4$ flux upscaling models.

**RC:** Climatic data: You seem to use also PAR data at 1 km spatial resolution (Table A1). Please add a short description of these data in the main text.

**AC:** Thank you for this comment. We have now added a clear description of the PAR dataset in the main text (after line 152), noting its spatial resolution, data source, and temporal downsampling approach.

**RC:** l161: you start describing Sentinel-2 data before the sentence about 10 m stack. Please reorganize the paragraph.

**AC:** We reorganized the paragraph in Section 2.2.3 so that the description of Sentinel-2 preprocessing now follows immediately after the introduction of the 10-m predictor stack. This improves clarity and logical flow.

**RC:** l162: Why 30 neighborhood for TPI? In general, you should test multiple neighborhood distances.

**AC:** The 30-m window provided the best balance between capturing microtopographic relief and avoiding excessive smoothing. We have now clarified this in the Methods (line 163) as well as in Table A1

**RC:** l190: Why NDVI and NDWI? Why not other indices such as NDMI? NDVI and NDWI have typically very high negative correlation.

**AC:** NDMI was not included because our 1 m orthomosaic contains only RGB + NIR bands and lacks the short-wave infrared (SWIR) channel required for NDMI computation. For Sentinel-2 data (10 m), we tested NDMI but found it highly collinear with NDWI and providing no improvement in model performance. NDVI and NDWI were therefore retained as the most interpretable and widely used vegetation and moisture indices for high-latitude ecosystems. Although the two indices are strongly negatively correlated (Spearman's $\rho$ = -0.93 at 1 m and -0.98 at 10 m resolution) due to their shared NIR component, they describe distinct ecological mechanisms: NDVI represents vegetation greenness and photosynthetic activity, while NDWI captures surface and canopy moisture. Including NDWI improved model performance ($\Delta AIC \approx 100$, $\Delta R2 \approx 0.01$), and the non-parametric models applied are robust to such predictor correlations. Retaining both indices allows a more complete representation of vegetation-moisture interactions characteristic of Arctic heterogeneous microtopography.

**RC:** l195: What do you mean by "sensitivity tests"?

**AC:** We agree that the original wording was unclear. We revised the text to explicitly describe what was tested and how performance was evaluated. Instead of "sensitivity tests," we now state that each additional predictor (CALU, InSAR, time-specific NDVI/NDWI) was included in separate model runs to assess whether it improved predictive skill ($R^2$, RMSE) compared to the main predictor set. This clarification is included in lines 195-199.

**RC:** l198: What do you mean by isolating "the effects of scale"?

**AC:** We clarified that only the spatial resolution (1 m vs. 10 m grid size) differs between the two datasets, while all other modelling inputs and processing steps remain identical. This ensures that differences in performance and outcomes can be attributed specifically to scale (lines 198-199).

**RC:** l215: "The overall workflow is summarized in Fig. 3. " The sentence can be deleted.

**AC:** We agree and have removed this sentence from the revised manuscript.

**RC:** l216: It would be good to describe already here that the climatic variables were spatially uniform over the study area.

**AC:** Thank you for this suggestion. We have now added a sentence in the revised Methods (Section 2.3.1) explicitly stating that AT, PAR, and TDD were varying only temporally, while all other predictors were spatially distributed.

**RC:** l217: Is there kind of double counting if some of the variables are first used for landscape classification and then again for the regression models together with the landscape classification. Is the classification needed as a predictor in the regression analyses?

**AC:** We appreciate the reviewer's thoughtful question. The landscape classification was included as a categorical predictor to represent vegetation and microtopographic heterogeneity that cannot be fully captured by continuous predictors such as NDVI, NDWI, or terrain indices. Although some of these variables were among those used to derive LC, the classification was based on a much broader set of spectral, texture, and topographic parameters (see Table A2). LC therefore summarises complex, multi-source information into discrete ecological units (e.g., sedge, tussock, or lichen patches) that reflect vegetation composition and hydrological conditions. Including LC thus provides complementary ecological information rather than redundant input, and models excluding LC performed less consistently across sites.

**RC:** l239: Did this analysis include temporally variable NDVI and NDWI? How about CALU and subsidence? Please state clearly that the analysis was done for both spatial resolutions.

**AC:** Thank you for raising this point. We clarified in the revised manuscript that CALU, InSAR subsidence, and temporally dynamic NDVI/NDWI were only tested at 10 m resolution, because they are not available at 1 m. We also explicitly stated that these variables were used only in separate exploratory model runs to evaluate their added predictive value, and therefore were not included in the main 1 m vs. 10 m resolution comparison. The revised text (after line 238) now clearly communicates which predictors were assessed at each spatial scale.

**RC:** l251: How was the differencing done? Was the 10 m maps resampled to 1 m spatial resolution first? What does "differences between model families" mean; differences between random forest and gradient boosting? What data was subtracted from what data?

**AC:** We thank the reviewer for pointing this out. We now explain that 1 m predictions were aggregated to 10 m spatial resolution before computing pixel-wise differences, ensuring direct comparability. We also clarify that "model-family differences" specifically refer to the subtraction of GBM minus RF predictions to evaluate structural uncertainty. These clarifications were added in the revised Methods (lines 251 – 253).

**RC:** l255: What does "tuned" mean here?

**AC:** We clarified that "tuned" refers to cross-validated hyperparameter optimisation. The sentence now explicitly states that variable importance was extracted from models after hyperparameter optimisation. This removes any ambiguity (Line 255).

**RC:** l273: You should test SAGA wetness index which spreads high wetness values to larger neighborhoods. It could produce spatially more coherent result than traditional TWI.

**AC:** We appreciate the suggestion. We agree that smoothed wetness metrics can reduce noise in some settings. However, we intentionally did not use the SAGA Wetness Index (SWI) in this study because it would compromise our resolution-controlled experiment. SWI expands and smoothes wet areas based on local topography, which would mask the very narrow wet features that drive $CH_4$ fluxes at 1 m resolution, and would alter the moisture patterns differently at 10 m due to the coarser DEM input. As a result, SWI would introduce changes unrelated to pixel size and therefore prevent us from isolating the true effect of spatial resolution on model behaviour. For these reasons, we kept TWI as a consistent and resolution-neutral moisture proxy across both scales.

**RC:** Table 2: What was N in the correlation analyses? How many temporal observations for NDVI and NDWI?

**AC:** We clarified this in the main text (line 285). All correlation analyses were performed using the full dataset of 13,384 chamber flux observations. Temporally dynamic NDVI and NDWI values were successfully extracted for every flux measurement (within a 10-day window), so sample size was identical for static and time-matched predictors.

**RC:** l298: "This also applies to the correlations reported in Table 2." This sentence could be deleted.

l301: This sentence does not continue anywhere.

**AC:** We agree that these sentences are not essential for interpretation and have removed them from the revised manuscript.

**RC:** l311: Similar flux stratification compared to what?

**AC:** We have reformulated the sentence

**RC:** l320: Overlay analysis of what?

**AC:** We updated the text (line 320) to state explicitly that the overlay compared our site-specific landscape classification with the CALU classification. This makes the purpose of the comparison clear.

**RC:** Figure 5: This is a good figure. It would be good to conduct a similar analysis between CALU and 1 m landscape classification.

**AC:** Thank you for this suggestion. We have now added the requested comparison using the 1 m classification, and the results are presented as Figure S2 in the Supplement. The text has been updated accordingly (after line 324) to state that the degree of cross-class mixing is similar at 1 m and 10 m resolution, indicating that the differences primarily reflect conceptual distinctions between classification schemes rather than grid size.

**RC:** l337: In the methods, you write that you emphasize RMSE in model comparison and here you state that you emphasize MAE. In reality, you seem to emphasize R2 a lot (or at least you report it). Please be consistent. Please state also how you calculated R2 in the methods section. Is it just squared correlation?

**AC:** We have revised the Methods section (lines 234-236) to clearly describe all three evaluation metrics and to ensure consistency with how results are presented. The text now specifies that $R^2$, RMSE, and MAE were all derived from cross-validated predictions using the yardstick package, with $R^2$ representing the coefficient of determination calculated from the correlation between observed and predicted values. We also clarified that $R^2$ and RMSE were the main metrics used for model evaluation, while MAE was included as an additional indicator of absolute error.

**RC:** l342: You seem to report inconsistently both normalized and unnormalized MAE and RMSE. Please be consistent. If you normalized these values, how did you conduct the normalization?

**AC:** Thank you for noticing this. We have removed the normalized values to ensure consistency across all reported metrics. The manuscript now reports only unnormalized MAE and RMSE, both derived directly from cross-validated predictions.

**RC:** Table 3: There are no bold values in the table despite caption claims it. What is the unit for MAE and RMSE? Can you include also normalized values?

**AC:** We have now added bold formatting to indicate the best-performing models as stated in the caption and included units for MAE and RMSE in both the table and caption. We decided not to include normalized values to maintain consistency with the unnormalized metrics reported throughout the manuscript.

**RC:** l366: Is there a need to refer to MAUP? Can you speculate this with a plainer language? Do you have any evidence that pixel aggregation is the reason for the model explanatory capacity behavior?

**AC:** We have changed the text accordingly (lines 366-376).

**RC:** l396-397: There seems to be little logic between the first and second part of this sentence.

**AC:** We have changed the text accordingly (lines 396-397).

**RC:** l423: Isn't this in contrast with the earlier claim that GBM can predict extreme values better?

**AC:** Thank you for noting this possible ambiguity. We have rephrased the sentence to clarify that both models predict a comparable overall flux range, while their main differences arise from the distribution of intermediate values rather than from absolute minima or maxima. GBM occasionally produces stronger negative extremes, which aligns with its higher sensitivity to localized sink conditions. This revision removes the apparent inconsistency and better reflects the intended meaning (lines 423 - 428).

**RC:** Figure 7: Can you have the chamber-measured fluxes in this figure? It would also be a good idea to compare the upscaled fluxes with the chamber-measured fluxes the text.

**AC:** We appreciate this suggestion. However, chamber measurements represent point-scale fluxes at specific microsites, whereas the upscaled flux maps in Fig. 7 depict spatially continuous mean fluxes for the entire area of interest. Because chamber plots occupy only a small fraction of the mapped surface and are unevenly distributed across land-cover types, directly overlaying them on the spatial maps would not provide a meaningful or representative comparison.

**RC:** l459: Please explain in the methods section how you normalized the importance values.

**AC:** We will clarified in the Methods section that variable importance scores were normalized within each model by dividing by the sum (or maximum) of all importance values, so that the resulting values range between 0 and 1 and are directly comparable across models and resolutions (after line 257).

**RC:** l462: Are the terrain measures correlating? Please give evidence, do not just speculate.

**AC:** We agree and have now added supporting evidence. A correlation matrix (Fig. B1) was added to the Supplement to quantify relationships among terrain predictors. The analysis shows moderate correlations between TPI and TWI (r = -0.38 at 1 m; -0.59 at 10 m) and between Slope and TWI (r = -0.14 at 1 m; -0.50 at 10 m), while Aspect is only weakly correlated with other topographic metrics (r < 0.2). Correlations between terrain, moisture, and vegetation indices remain low (r < 0.3). These results confirm that some intercorrelation exists among topographic predictors, consistent with their shared DEM origin, but that it does not extend strongly to other predictor groups.

**RC:** l476: You could cite more the remote sensing-based WT studies here.

**AC:** Thank you for the suggestion. We have added an additional reference that explicitly links remote sensing indices to water-table estimation in peatlands (Meingast et al., 2014) to strengthen this statement (Line 476).

**RC:** Figure 8: The division into the categories seems a bit arbitrary. Why TWI is not under topography? Should you have instead topography, spectral, meteorological and landscape categories here?

**AC:** We agree that alternative groupings such as "topographic, spectral, meteorological, and landscape" could be used. We chose a process-based structure to emphasise the functional roles of predictors in $CH_4$ regulation rather than their data origin. For instance, TWI was placed under "Hydrology/Moisture" because, although derived from topography, it quantifies surface and subsurface water accumulation - a key driver of methanogenesis. NDWI was grouped there for the same reason. This approach highlights how each predictor relates to $CH_4$-controlling processes instead of its measurement source.

**RC:** Feature importance analysis: it would be good to have the analysis also for subsidence, CALU and temporal NDVI/NDWI.

**AC:** We thank the reviewer for this valuable suggestion. We have now extended the feature importance analysis to include models incorporating subsidence, CALU, and temporally matched NDVI/NDWI indices. The new results are provided in the Supplementary Information (Text S3, Fig. S3, Fig. S4), where we describe their influence on $CH_4$ flux prediction.

**RC:** Limitations: Can you integrate the 3.5 into the earlier sections? It feels a bit odd to have a section with only a couple of sentences.

**AC:** We agree and have integrated the limitation points into the earlier relevant sections of the Results and Discussion. The discussion on data imbalance and missing soil variables is now mentioned in Sections 3.1 (data coverage and CALU classes) and 3.2 (model evaluation and missing predictors). The standalone Section 3.5 has been removed.

**RC:** l495: Can you quantify and describe the unbalanced sampling more?

**AC:** We have now quantified the sampling imbalance directly in the text, specifying that tall-shrub areas represent < 2 % of flux points while covering ~25–30 % of the mapped area. This clarifies which landscape types were underrepresented and why this may affect flux variability.

**RC:** l497: "were not included"; as predictors?

**AC:** Yes, we refer to soil temperature and soil moisture as potential predictors. These variables were not included in the current models because spatially consistent data were not available at the time of analysis. This clarification has been added to the text.

**RC:** Conclusions: Please rewrite the conclusion section. Give a brief overview of the study aims and then answer to each four research questions one-by-one.

**AC:** We thank the reviewer for this suggestion. The conclusion section has been fully revised to begin with a concise overview of the study aims and to explicitly address each of the four research questions in order.

**RC:** l508: How the targeted sensitivity analyses could be done?

**AC:** We clarified this in the revised text. Specifically, we now specify that robustness can be tested through targeted sensitivity analyses, including tests with modified predictor sets, varied hyperparameters, and bootstrapped subsampling to evaluate the stability of variable importance and model performance. This provides a clear framework for how model stability could be quantitatively assessed (after line 508).

**RC:** l514: Acronyms such as RF-1 m are a bit difficult for the conclusions section.

**AC:** We revised the phrasing for clarity, separating model acronyms and resolution indicators (e.g., "RF at 1 m resolution" instead of "RF-1 m") at line 514.

**RC:** l519: Should it be "coarser resolution models can outperform ultra-high spatial resolution models"?

**AC:** We appreciate the comment. In our terminology, 1 m represents ultra-high and 10 m high spatial resolution. The sentence intentionally compares these two levels, as 10 m models (high-resolution) occasionally outperformed 1 m models (ultra-high-resolution) when temporally dynamic predictors were included. We clarified this wording in the text to avoid confusion (line 519).

**RC:** l520 and generalizability: Are you writing only about spatial resolution here?

**AC:** We clarified that both terms refer to spatial aspects (line 520).

**RC:** Table A1: Should the resolution column be entitled "spatial resolution"? Text about slope and aspect could be shorter. The reference to Wilson & Gallant feels a bit odd as your DTM is not from their work.

**AC:** The table was updated accordingly.

**RC:** Table A2: Please explain the abbreviations in the caption. Spatial resolution for individual bands is 10 m; were they not calculated for 1 m spatial resolution? Did you use SWIR and RE bands for Sentinel-2? You could include "spatial" in the resolution column header for this table also.

**AC:** The table caption was revised to define all abbreviations and clarify that indices were derived from Sentinel-2 (10 m) and drone imagery (1 m) using the visible and near-infrared bands (Blue, Green, Red,

NIR). SWIR and red-edge bands were not used to maintain consistency between the 1 m and 10 m classifications.

**RC:** l563: Please report also class-specific accuracies. You could also add a confusion matrix.

**AC:** We have now added class-specific user accuracies for all landscape classes and included confusion matrices for both 1 m and 10 m classifications (Table S1, Supplementary Information). Because the validation dataset included only 28 points ( 20 % of 140 training sites), several classes had ≤ 2 test points (e.g. dwarf shrub, sedge), making class-specific accuracies statistically unstable. The main text was updated to reference these additions (Section 2.2.3).

**RC:** Figure B1: The caption should be below the figure. How was the difference calculated? What minus what? Can you provide also the difference maps? Are middle and right-hand columns needed? They are a bit difficult to digest and they are not referred to in the main text.

**AC:** The figure was renamed to Figure B2 and the caption revised to explicitly describe the calculation of each difference (RF – GBM and 1 m – 10 m) and the direction of resampling (10 m downscaled to 1 m and 1 m aggregated to 10 m). The caption was also moved below the figure. The underlying difference maps are available in the project's supplementary repository (Zotero dataset) to allow overlay and detailed inspection with other layers. A short note referencing this availability was added to the main text in Section 3.3 (lines 445-446).

**RC:**

l27: microtopography is -> microtopography was

l28: sub-metre: you do not have any sub-meter predictors as the finer tested spatial resolution was 1 m. You write also in other parts about sub-meter resolution. Please be consistent about writing of 1 m spatial resolution (not higher than that).

l74: meter or metre: You seem to use mostly British spelling but not consistently.

l296: Should this paragraph be above the previous paragraph?

l264-267: There is little information value in this paragraph. Consider deleting.

l315: Delete "the" from the end of the line.

l355-356: Please put these two paragraphs together.

**AC:** Thank you for these helpful suggestions. We corrected the wording and ensured consistent use of 1 m resolution terminology throughout the manuscript. Spelling has been standardised to British English (e.g., metre). We also removed the indicated paragraph with limited value, merged or reordered the affected paragraphs for improved flow, and updated all line references in the revised version.

**Reviewer 2**

**RC:** Representativeness of sites: It is unclear whether the sample is representative of the entire study area. The data appear to be concentrated in a few locations, which is understandable given the logistical challenges of conducting fieldwork across large wetland areas. Nevertheless, this aspect should be described in more detail in the 'Methods' and 'Results' sections; a single sentence in the 'Limitations' section is insufficient.

**AC:** We agree that this aspect required a clearer description. As also suggested by Reviewer 1, we have substantially expanded the explanation of spatial representativeness. Section 2.2.1 now quantifies the number and distribution of chamber sites and flux observations across dominant land-cover types. These include tussock tundra, dwarf-shrub tundra, lichen-dominated uplands, and sedge wetlands, together covering the full wet-to-dry gradient of the study area. Figure 1 and a new Supplementary Table S2 now illustrate this coverage and confirm that all key microtopographic units controlling CH4 fluxes were represented.

Changed in manuscript: New text added in Section 2.2.1 (after original L143) and Table S2 added to Supplement.

**RC:** A more detailed discussion of site representativeness would be beneficial, including the number of sites per land cover type, and the number of measurements per site. In Figure 4, n appears to refer to the total number of measurements, but it would also be useful to indicate the number of sites per land cover category there, as well as in Tables B1 and B2, or somewhere else. This would facilitate discussion of this limitation; e.g., the text mentions that the 'wetland, permanent' class includes only one site which.

**AC:** The total number of measurements per land-cover class is already shown at the top of Figure 4. To improve transparency, we have now added the number of sites per class directly in the figure caption and included both site and measurement counts in Table B1.

**RC:** Additionally, are sites weighted differently in the model? For example, automatic chambers likely produce more measurements than manual ones — do these sites then have a greater influence on model training? How do you account for potential site-level overfitting? Did you consider using a leave-one-site-out cross-validation approach to assess the robustness of the model in predicting new areas where no data was used for training?

**AC:** All chamber measurements were treated equally in model training, without explicit site-level weighting. Automatic chambers indeed produced more frequent measurements and therefore contributed a larger number of samples, reflecting their higher temporal resolution. Automatic and manual chambers were deployed in different land-cover types, except for one mixed class (class dwarf shrubs), where both were present. Because overlap was limited, the larger number of automatic-chamber observations did not substantially bias model training toward specific surface types. To prevent overfitting to individual sites, we applied grouped five-fold cross-validation using Site as the grouping variable, ensuring that all data from a given site were contained entirely in either the training or testing subset, but never in both. This setup prevents data leakage and provides a robust assessment of model transferability to unseen sites. We also performed leave-one-site-out (LOSO) cross-validation, which produced results similar to the grouped CV; therefore, it was not used as a separate evaluation.

**RC:** Finally, could the differences between the two models (particularly at 10 m in Figure 6) over specific areas/LC types be explained by a lack of training data in these areas/LC types?

**AC:** We agree that limited data coverage in specific land-cover types contributes to part of the model disagreement, especially at 10 m resolution. Tall-shrub and complex wetland zones were sampled less intensively, mainly due to access constraints, and these classes show higher model uncertainty and stronger divergence between RF and GBM. However, the differences cannot be attributed to data gaps alone: GBM tends to amplify local extremes, whereas RF smoothes them, leading to systematic contrasts in sparsely sampled or highly heterogeneous surfaces.

Changed in manuscript: lines 453-455.

**RC:** Mismatch between data input and resolution effect :

The comparison between the 1 m and 10 m datasets is particularly interesting, as it reveals the differences in the two approaches with commonly used input data at these resolutions. However, this comparison potentially combines two effects: one related to the resolution itself (average over a larger area), and another related to potential differences in the data sources themselves (e.g., different acquisition date/time, different sensors...). Have you attempted to separate these two influences? One way to do this would be to aggregate the 1 m product to 10 m and apply the same workflow (e.g. for land cover, use the dominant vegetation type within each 10 m grid cell and take the mean for the other variables). This could help to isolate the effect of the resolution from that of the different data sources.

Otherwise it would be useful to discuss this somewhere, and include a comparison of the datasets used as is done for the comparison with CALU (Figure 5), but for the two datasets at different resolutions (as is partly done for land cover in Figure 2, where important differences can be seen).

**AC:** We agree that the comparison between 1 m and 10 m models may mix the effects of spatial resolution and data source. To isolate the influence of spatial resolution alone, we aggregated the 1 m input data to 10 m resolution using the same workflow applied to the Sentinel- and ArcticDEM-based 10 m dataset. Continuous predictors were averaged within each 10 m grid cell, and categorical variables were assigned based on the majority class. The resulting "10 m from 1 m" dataset was then used to train and evaluate all models using the same hyperparameter configurations and cross-validation setup as in the main analysis (after original lines 238, Methods)

Model performance based on the aggregated 1 m data was very similar to that of the original 10 m models, with only small differences across algorithms. GBM and SVR slightly improved after aggregation, RF performed marginally worse, and GAM remained nearly unchanged (we added results into SI: Text S4, Fig. S4, Fig. S5). These results indicate that the differences between the 1 m and 10 m models presented in the main text are primarily driven by spatial resolution rather than by differences in the underlying sensor-derived input data.

**RC:** The mix of spatial and interannual analysis is somewhat confusing. It is unclear how the spatial and interannual components are distinguished in the study. It is not always obvious whether the analysis is spatial, temporal, or a combination of both. Although the study appears to be mainly spatial, with a single-month focus on July, it also uses temporally varying predictors only (AT, PAR and TDD over six years). Clarifying this in the text and figures (methods and results) would improve readability. The time-varying inputs are difficult to understand from the main text: which variables are dynamic and at what resolution? (See the comment about the data section below.)

**AC:** We have now added a sentence in the revised Methods (Section 2.3.1) explicitly stating that AT, PAR, and TDD were varying only temporally, while all other predictors were spatially distributed. We added a clarifying sentence in the Results section (line 389) explaining that interannual variation in predicted $CH_4$ fluxes arises from interactions between static spatial predictors and varying meteorological conditions, rather than from spatial changes in surface properties.

**RC:** Spatial and temporal accuracy should be discussed separately in the 'Results' section, or more explanations should be provided. For example, spatial correlations (mean flux per site) and temporal correlations (time series at individual sites) could be reported separately in Tables 2 and 3 to disentangle these effects and avoid sites with potential larger amounts of data dominating the analysis compared to sites with smaller amounts of data. A panel like 7B could be used to directly compare model predictions with measurements at the sites, providing a clearer assessment of spatial and temporal performance.

**AC:** We agree that, in principle, separating spatial and temporal accuracy could provide additional insights. However, in our dataset, the temporal and spatial dimensions are strongly confounded: most sites were measured in different years, and only a small subset was revisited under comparable environmental conditions. This makes it impossible to disentangle spatial from temporal variability without introducing artificial biases. For example, "leave-one-year-out" or "leave-one-site-out" validation would

simultaneously exclude unique combinations of both site characteristics and meteorological conditions, leading to unrealistically low or unstable skill estimates that reflect data gaps rather than true model limitations. Instead, we report the results of grouped cross-validation by site and by year (Text S4, Fig. S7), which show the expected decrease in R² and increase in RMSE when models are tested beyond their training conditions. These values reflect the structure of the dataset rather than model weakness: most sites were not remeasured across years under comparable local conditions. Therefore, a further separation into "spatial" and "temporal" accuracy would produce misleading metrics. We include the grouped CV results in the Supplement for transparency but do not expand them into separate spatial or temporal tables.

**RC:** The data section of the Methods section needs to be restructured and expanded. Section 2.2.3 (and the Materials and Methods section more generally) should be reorganised, as it is currently difficult for the reader to determine which datasets are used at 1 m, which at 10 m, and which at both resolutions. For instance, the text initially focuses on 1 m data, but then abruptly shifts to Sentinel-2 (presumably 10 m) before describing the 10 m products. References to 30 m window data are confusing and require explanation. The temporal dimension of each variable is unclear too. While some of this information appears in Table A1, Figure 3 and lines 240–247, the description remains fragmented. Providing a summary table that explicitly lists the ten variables used for each resolution, their data source, spatial resolution (1 m, 10 m or constant) and whether they are static or dynamic would certainly help the reader.

**AC:** The requested information is already summarized in Table A1, which lists all ten predictor variables together with their source, spatial resolution, and temporal type (static / dynamic). To improve clarity, we have restructured Section 2.2.3 to clearly separate 1 m UAV-based, 10 m Sentinel-based, and spatially uniform meteorological variables. We also added explicit reference to Table A1 at the beginning of the subsection  The 30-m window provided the best balance between capturing microtopographic relief and avoiding excessive smoothing. We have now clarified this in the Methods (line 163) as well as in Table A1.

**RC:** Data processing procedures should also be described in more detail in Section 2.2.3 or in a dedicated section. For instance, how were Sentinel-2 data cleaned or filtered? Were cloud-free conditions explicitly selected for the time-varying Sentinel-2 indices? This is implied by lines 278–281, but stating this explicitly in the 'Remotely Sensed Data' section would improve transparency. Overall, providing a clearer and more detailed description of the data pre-processing and management would strengthen the reproducibility of the study.

**AC:** We now explicitly describe the Sentinel-2 preprocessing steps in Section 2.2.3 to ensure full transparency and reproducibility. All Sentinel-2 Level-2A scenes were cloud-, shadow-, and snow-masked using the QA60 and Fmask layers before any compositing or index extraction. Only cloud-free scenes with <10 % overall cloud cover were used. The July–August 2018 composite was generated from six completely cloud-free scenes; thus, no cloudy pixels entered the mosaic. For the time-specific NDVI and NDWI datasets, we extracted the closest cloud-free Sentinel-2 image within ±10 days of each chamber flux measurement, without temporal averaging and including only pixels that fully passed cloud masking. All preprocessing was implemented in Google Earth Engine, and the processing scripts are available in the public project repository.

Changed in manuscript: Clarifications added in Section 2.2.3 (Lines 165–172).

**RC:**  for the chamber data, management should also been specified. How is chamber data managed spatially? How are fluxes aggregated at 1 m or 10 m resolution — do you take the mean of all chambers within each 1×1 m or 10×10 m pixel? You mention PAR and other variables measured at chamber sites. Are these used here ? Providing this information is essential for understanding how point-scale observations are scaled to the model resolutions. Additionally, since chamber flux measurements are known to be highly variable, it would be useful to specify in the methods section whether each flux observation corresponds to a single or repeated measurement.

**AC:** We now provide a clearer description of how chamber flux data were linked with the predictor layers and how spatial variability was treated. Each chamber flux measurement was linked to the predictor grids via direct extraction from the raster cell covering the chamber footprint, without spatial buffering or neighbourhood smoothing. No averaging was applied at either 1 m or 10 m resolution. Even when multiple chambers or repeated measurements fell within the same grid cell, all flux observations were retained as separate records to preserve sub-pixel heterogeneity in vegetation, moisture, and microtopography. All $CH_4$ flux measurements represent individual chamber events. These details and clarifications have been added in Section 2.2.3 Remotely sensed data.

**RC:** I do not understand Figure 7A. According to the caption, it should show monthly estimates averaged over the entire area, but the large number of points is confusing.

**AC:** Figure 7A indeed shows monthly $CH_4$ flux estimates for individual pixels across the study area, not a single aggregated mean. The large number of points reflects spatial variability within the domain. We clarified this in the caption and indicate that each point corresponds to a pixel-level monthly mean to improve readability.

**RC:** Have you considered using a "leave-one-site-out" or "leave-one-year-out" cross-validation (e.g. training on the first three years and predicting the last year) ? This could enable assessing how well these models can predict pixels/sites or or time for which no data was used in the training process, as well as the uncertainties related to each model training, which are not really discussed here.

**AC:** We performed grouped cross-validation using both Site and Year as grouping variables to evaluate model transferability beyond the calibration domain. In the grouped-by-site CV, all data from a given site were held out entirely in the test set, while in the grouped-by-year CV, all measurements from a given year were excluded during training. These setups are equivalent to leave-one-site-out and leave-one-year-out approaches but implemented with balanced folds to ensure stable statistics.
As expected, $R^2$ decreased and RMSE increased compared to standard v-fold CV, because environmental conditions varied strongly between sites and years. Many sites were not revisited in all years, and specific landscape classes (ex., drier) were sampled predominantly in earlier campaigns (2019-2021), while wetter surfaces were measured mainly in later years (2022-2024). Thus, holding out entire years or sites effectively removed key combinations of soil moisture, vegetation, and temperature from the training data, making it difficult for models to predict fluxes under unseen environmental contexts. Nevertheless, the models retained the correct direction of $CH_4$ fluxes (high fluxes at wet sites, near-zero or negative at dry ones), indicating a consistent mechanistic response despite reduced fit.
These results underline a critical need for repeated measurements under comparable local conditions across years to better constrain interannual model generalization.
For clarity, we summarize these diagnostics in the Supporting Information (Text S4, Fig. S7) and briefly discuss them in the revised Results sections (line 376).

**RC:** The emphasis of what are the main findings differs between the abstract, the main text and the conclusion. Please clarify. For example, the substantial difference in emission estimates between the RF and GBM models (519 vs. 99 mg $CH_4$ m$^{-2}$) is emphasised in the abstract, but this is not discussed in the same way in the main text or conclusion. I think this should also be highlighted in the conclusion. Conversely, the potential advantages of the 10 m estimates, which are mentioned in the conclusion (in particular lines 519–520), are not emphasised in the abstract. This important point should probably be included in the abstract to provide a more coherent overall message.

**AC**: We revised the Abstract, Results (Section 3.3), and Conclusion to ensure a consistent interpretation of model and resolution effects throughout the manuscript. In the abstract, we now clearly mention both (i) the strong contrast in mean $CH_4$ fluxes between RF and GBM at 1 m resolution and (ii) the convergence of these estimates at 10 m resolution due to spatial aggregation. In the main text (Section 3.3), we expanded the interpretation to explain that the fivefold difference at 1 m reflects structural uncertainty arising from

algorithm selection, while the reduced discrepancy at 10 m results from the smoothing of microtopographic extremes. In the conclusion, we added that although the two algorithms differ substantially at the highest spatial resolution, this divergence diminishes once predictors are aggregated, indicating that coarser grids stabilize landscape-scale $CH_4$ budgets.

These revisions make the main findings coherent across all sections and explicitly connect the implications of both model choice and spatial resolution.

Changed in manuscript: Abstract, lines 31- 33;  Section 3.3, lines 409-412; Conclusion, lines 508-509

**RC:** lines 234-236 «Root-mean-square error (RMSE) between measured and predicted CH4 fluxes was the primary comparison metric because it penalizes large deviations more strongly than a mean-absolute error (Chai & Draxler 2014).» seem contradictory with lines lines 336-337 «Due to the relatively low mean CH4 flux across all sites (0.102 mg CH4 m-2 h-1), the emphasis of our model evaluation was placed on absolute errors (MAE) rather than the fraction of explained variance (Table 2).». Please clarify.

**AC:**  We agree that the phrasing was inconsistent. To ensure clarity, we removed the sentence that emphasized MAE in the Results section. RMSE remains the main evaluation metric because it penalizes large deviations more strongly, while MAE and $R^2$ are reported in Table 2 as complementary indicators of model error magnitude and explained variance.

**RC:** There is generally little discussion about the application, relevance, and limitations of this research within a broader context. A discussion of its potential and limitations for broader application would strengthen the manuscript. Below are some suggestions for further discussion.

**AC:** We agree and have clarified these aspects throughout the manuscript. Section 2.1 (after original line 125) now links the study site to broader Arctic tundra settings, Section 2.2.3 (before original line 194) includes a note on scalability of 10 m versus 1 m inputs, and the Conclusion now summarises the workflow's transferability, key limitations, and recommendations for wider application (before original line 521).

**RC:**  Seasonal/monthly variations are not presented in this manuscript, but they should be discussed in the 'Discussion' or 'Limitations' sections.

**AC:** We agree and have clarified this in the Conclusion. The text now explicitly notes that our analysis is limited to July fluxes and outlines how extending the workflow across seasons and additional sites would help assess seasonal dynamics and strengthen the generality of our findings.

**RC:**  The study uses chamber measurements focusing on soil and short vegetation emissions. How could this approach be generalized to other ecosystems, particularly forested or tree-dominated systems? Would the method differ if measurements were taken at the ecosystem scale such as Eddy Covariance that average emissions, but also take vegetation emissions (which can be very substantial part of the emissions) into account? Could comparisons with flux estimates from larger-scale approaches — for example, eddy covariance tower or aircraft-based eddy covariance campaigns over the same areas of interest (e.g., Shaw et al., 2022, https://doi.org/10.1029/2021GB007261; Sayres et al., 2017, https://doi.org/10.5194/acp-17-8619-2017) provide valuable evaluation of the upscaling methods and help assess their robustness across spatial scales ?

**AC:** We appreciate the reviewer's suggestion and agree that comparisons with ecosystem-scale flux measurements would provide valuable context. At Trail Valley Creek, the main EC tower footprint covers only a subset of the heterogeneous tundra landscape, which limits the representativeness of a direct comparison. Moreover, EC observations integrate fluxes over heterogeneous surfaces and therefore do not resolve patch-level variability that is central to our upscaling approach. However, if EC datasets can be decomposed into contributions from different surface types, then such data could be used to evaluate and

refine our models across spatial scales. Regarding other ecosystems, such as forests, extending this workflow would require more representative chamber datasets and adjusted predictor sets. In forested systems, canopy structure often obscures the microtopographic and moisture gradients detectable by remote sensing, complicating direct transferability. Overall, additional data tailored to each ecosystem type would be needed to ensure robust model application.

**RC:** Linking these fine-scale results to broader $CH_4$ budgets, which are usually estimated at coarser resolutions, raises questions about scalability. Why were only 1 m and 10 m resolutions considered? Would other coarser scales (50 m, 100 m, 1 km) be relevant ? The comparisons mentioned in lines 436–440 refer to models run at 0.25–0.5°. Are these results directly comparable? How could your findings be used in larger-scale budgets?

**AC:** We thank the reviewer for raising this important point. We limited the analysis to 1 m and 10 m resolutions because these are the only scales where chamber observations remain spatially representative and directly usable for model training and validation. At coarser resolutions ($\geq$ 50–100 m), single pixels would contain multiple microhabitats, and most chamber measurements would collapse into one or two pixels, making direct scaling and validation unreliable. The 10 m scale, however, is directly transferable to regional mapping, as Sentinel-2 and ArcticDEM provide continuous Arctic coverage at this resolution. Our results therefore bridge the process scale of chamber observations with the resolution domain used in regional models and inventories. Our 1–10 m maps can support larger-scale budgets in two ways: (1) as per-pixel validation for the limited set of pixels that directly contain chamber measurements, and (2) as high-resolution input for sub-grid upscaling, where 10 m maps are aggregated area-weighted to coarser grids while preserving the fractional cover of wet and dry micro-sites. This helps coarse models represent unresolved fine-scale variability more realistically.

**RC:** Line 175: Could you please clarify how water pixels are defined? Could a wetland pixel be masked?

**AC:** We changed lines 175-176 to explain this: "Water pixels were masked prior to the classification using a threshold of NDWI > 0 and manually checked against the drone orthomosaic to ensure the exclusion of ponds and streams."

**RC:** lines 173-175 : «At 1 m resolution, the Tall shrubs + trees class was merged with Dwarf shrubs due to the absence of chamber measurements within that class, resulting in five effective classes at 1 m and six at 10 m.» It is unclear why there are six classes at 10 m. The point(s) that was (were) classified tall schrubs + tree at 1 m was not in a pixel classified in this category at 10 m ? Please clarify.

**AC:** Both the 1 m and 10 m land-cover maps include the same six classes. However, for model training at 1 m resolution, the Tall shrubs + trees class was merged with Dwarf shrubs because no chamber measurements overlapped pixels of that class in the flux dataset. Therefore, while six classes exist in both maps, only five were effectively used in the 1 m model due to the lack of in-situ flux observations within the Tall shrubs + trees category. At 10 m resolution, this class was retained because it is present in both the map and the corresponding training data. We explained it in text (lines 173-175).

**RC:** lines 78-79 «Moreover, ultra-high resolution can even introduce noise and not necessarily lead to a better representation of environmental conditions (Riihimäki et al., 2021).» seems contridactory with lines 70-75. Please clarify this sentence.

**AC:** The two sentences refer to different aspects of spatial resolution. We revised the text to make this distinction explicit (lines 70–79).

---

## Referee Report (RR1)

Overall, I'm happy with both the authors' comments and their efforts.

However, there is one important point that I don't fully understand, but perhaps I'm missing something here.

This relates to my questions about site overfitting. The authors responded to my concerns that they were using five-fold cross-validation grouping data per site (is it really the case in the general model optimisation setup ?). Five-fold cross-validation is then supposed to prevent site overfitting, as the data of one site is either used for training or evaluation, but not both. But they also show some tests they made for CV-Site in the supplementary material. I am then wondering why the Site-CV results (Text S4 and Figure 7) are so different from the five-fold CV results, as both methods should avoid site overfitting?

I understood that 'Grouped by Year' results were not really meaningful here. However, the results shown here for 'Grouped by Site' would indicate very poor model reliability. This would change the results of the study and call into question the ability of the models to reproduce fluxes and to be upscalled over the area.

The two RC/RA items related to this are listed below:

*RC: Additionally, are sites weighted differently in the model? For example, automatic chambers likely produce more measurements than manual ones — do these sites then have a greater influence on model training? How do you account for potential site-level overfitting? Did you consider using a leave-one-site-out cross-validation approach to assess the robustness of the model in predicting new areas where no data was used for training?*

*AC: All chamber measurements were treated equally in model training, without explicit site-level weighting. Automatic chambers indeed produced more frequent measurements and therefore contributed a*
*larger number of samples, reflecting their higher temporal resolution. Automatic and manual chambers were deployed in different land-cover types, except for one mixed class (class dwarf shrubs), where both were present. Because overlap was limited, the larger number of automatic-chamber observations did not substantially bias model training toward specific surface types. To prevent overfitting to individual sites, we applied grouped five-fold cross-validation using Site as the grouping variable, ensuring that all data from a given site were contained entirely in either the training or testing subset, but never in both. This setup prevents data leakage and provides a robust assessment of model transferability to unseen sites. We also performed leave-one-site-out (LOSO) cross-validation, which produced results similar to the grouped CV; therefore, it was not used as a separate evaluation.*
*+*
*RC: Have you considered using a "leave-one-site-out" or "leave-one-year-out" cross-validation (e.g. training on the first three years and predicting the last year) ? This could enable assessing how well these models can predict pixels/sites or or time for which no data was used in the training process, as well as the uncertainties related to each model training, which are not really discussed here.*

*AC: We performed grouped cross-validation using both Site and Year as grouping variables to evaluate model transferability beyond the calibration domain. In the grouped-by-site CV, all data from a given site were held out entirely in the test set, while in the grouped-by-year CV, all measurements from a given year were excluded during training. These setups are equivalent to leave-one-site-out and leave-one-year-out approaches but implemented with balanced folds to ensure stable statistics. As expected, R2 decreased and RMSE increased compared to standard v-fold CV, because environmental*
*conditions varied strongly between sites and years. Many sites were not revisited in all years, and specific landscape classes (ex., drier) were sampled predominantly in earlier campaigns (2019-2021), while wetter surfaces were*

*measured mainly in later years (2022-2024). Thus, holding out entire years or sites effectively removed key combinations of soil moisture, vegetation, and temperature from the training data, making it difficult for models to predict fluxes under unseen environmental contexts. Nevertheless, the models retained the correct direction of CH4 fluxes (high fluxes at wet sites, near-zero or negative at dry ones), indicating a consistent mechanistic response despite reduced fit. These results underline a critical need for repeated measurements under comparable local conditions across years to better constrain interannual model generalization. For clarity, we summarize these diagnostics in the Supporting Information (Text S4, Fig. S7) and briefly discuss them in the revised Results sections (line 376).*

*And the Figure S7 :*

[Figure]

**Figure S7. Model performance ($R^2$ vs. RMSE) for Random Forest (gray) and Gradient Boosting Machine (green) models under different cross-validation schemes (Standard CV = squares, Grouped by Year = triangles, Grouped by Site = circles) at 1 m and 10 m resolutions. Points show mean performance across folds; whiskers show standard deviation. Lower RMSE and higher $R^2$ indicate better performance.**

*citation of Text S4 : "Under a standard five-fold CV, models achieve high accuracy (R2 = 0.7-0.75, RMSE ≈ 0.06-0.07 for both 1 m and 10 m resolutions). In contrast, grouped-by-year and grouped-by-site CV produce much lower R2 values (typically 0.1-0.2) and larger RMSE (0.15-0.4)."*

Minor comment :

I'm not sure Figure's titles have been changed ?
perhaps a bug ?

*RC: A more detailed discussion of site representativeness would be beneficial, including the number of sites per land cover type, and the number of measurements per site. In Figure 4, n appears to refer to the total number of*

*measurements, but it would also be useful to indicate the number of sites per land cover category there, as well as in Tables B1 and B2, or somewhere else. This would facilitate discussion of this limitation; e.g., the text mentions that the 'wetland, permanent' class includes only one site which.*

*AC: The total number of measurements per land-cover class is already shown at the top of Figure 4. To improve transparency, we have now added the number of sites per class directly in the figure caption and included both site and measurement counts in Table B1.*

--> I don't think the changes worked in the figure caption.

*RC: I do not understand Figure 7A. According to the caption, it should show monthly estimates averaged over the entire area, but the large number of points is confusing.*

*AC: Figure 7A indeed shows monthly CH4 flux estimates for individual pixels across the study area, not a single aggregated mean. The large number of points reflects spatial variability within the domain. We clarified this in the caption and indicate that each point corresponds to a pixel-level monthly mean to improve readability.*

--> I think then "averaged over the entire area of interest" should be deleted.

---

## Author Response (AR2)

**Response to the Editor**

Dear Paul,

Thank you for your decision letter and for drawing attention to the interpretation of the grouped-by-Site cross-validation results. In our accompanying response to the referee, we provide a detailed description of how we clarified the model validation section in the manuscript.

At Trail Valley Creek, the landscape is strongly heterogeneous: some Sites (measurement locations) represent common combinations of vegetation, moisture and microtopography, whereas others capture rare or nearly unique conditions. When we perform grouped-by-Site cross-validation and hold out all data from a given Site, the environmental space occupied by that Site (for example, NDWI, TWI, landscape classification, microtopography) can be largely absent from the remaining training data. In such cases, the models are forced to extrapolate to combinations of predictors that have no close analogue in the training set, and it is therefore expected that $R^2$ values drop to ~0.1 - 0.2 in this very conservative scenario.

By contrast, the actual prediction task in this paper is to upscale $CH_4$ fluxes within the Trail Valley Creek area, across pixels that mostly fall inside the joint environmental space spanned by all chamber sites. For this within-domain setting, standard stratified k-fold cross-validation across individual measurements provides the relevant estimate of model performance, and those results show that RF and GBM capture spatial and temporal variability in $CH_4$ fluxes well. The grouped-by-Site analysis should therefore be interpreted as revealing the limits imposed by the small number of measurement locations in rare habitat types and the strong surface heterogeneity, rather than as evidence that the models are fundamentally unreliable.

We hope this clarifies why the grouped-by-Site results are not in conflict with our main conclusions and why the upscaling remains appropriate for the defined scope of the study, namely the heterogeneous surface at Trail Valley Creek.

To further align the body of the manuscript with the Abstract and highlight the utility of key remotely sensed variables, we have also added a sentence to the Abstract discussing the support offered by seasonal subsidence derived from remote sensing. This variable reflects important moisture gradients and shows high potential for improving $CH_4$ upscaling.

Sincerely,

Kseniia Ivanova

on behalf of all co-authors

**Response to Referee**

**RC:** Overall, I'm happy with both the authors' comments and their efforts. However, there is one important point that I don't fully understand, but perhaps I'm missing something here. This relates to my questions about site overfitting. The authors responded to my concerns that they were using five-fold cross-validation grouping data per site (is it really the case in the general model optimisation setup ?). Five-fold cross-validation is then supposed to prevent site overfitting, as the data of one site is either used for training or evaluation, but not both. But they also show some tests they made for CV-Site in the supplementary material. I am then wondering why the Site-CV results (Text S4 and Figure 7) are so different from the five-fold CV results, as both methods should avoid site overfitting? I understood that 'Grouped by Year' results were not really meaningful here. However, the results shown here for 'Grouped by Site' would indicate very poor model reliability. This would change the results of the study and call into question the ability of the models to reproduce fluxes and to be upscalled over the area.

**AC:** We apologise if our earlier reply created confusion about how cross-validation was implemented in the main analysis. Our previous wording may have suggested that the general model optimisation and all reported performance metrics were based on grouped-by-Site cross-validation, which is not the case.

In the main analysis, model tuning and performance assessment for the general models are based on standard stratified k-fold cross-validation across individual measurements, without grouping by Site or Year. All $R^2$, RMSE and MAE values reported in Tables 2 and 3 come from out-of-fold predictions of this standard k-fold CV. This setup corresponds to the prediction task we address in the paper: upscaling $CH_4$ fluxes within the Trail Valley Creek wetland complex, across pixels that mostly fall inside the joint environmental space spanned by all chamber sites.

By contrast, the grouped-by-Site and grouped-by-Year cross-validation runs shown in Text S4 and Fig. S7 are additional, deliberately conservative tests and were not used for model optimisation or for the main performance metrics. At Trail Valley Creek, Sites are measurement locations within heterogeneous area, and some Sites represent rare or nearly unique combinations of vegetation, moisture and microtopography. When we hold out all data from such a Site in grouped-by-Site CV, these conditions (for example, NDVI, NDWI, TWI) can be largely absent from the remaining training data. The models are then forced to extrapolate to combinations of predictors that have no close analogue in the training set, and in this strict setting it is expected that $R^2$ values drop to ~0.1–0.2.

We therefore interpret the grouped-by-Site CV results not as evidence that the general models are fundamentally unreliable, but as a stress test that reveals (i) how strongly performance deteriorates when entire measurement locations with rare combinations of conditions are removed, and (ii) how limited replication in rare habitat types constrains spatial transferability. For the within-domain upscaling task considered in this paper, the standard stratified k-fold cross-validation provides the relevant measure of model performance, and under this

evaluation RF and GBM reproduce the spatial and temporal variability in $CH_4$ fluxes reasonably well.

To avoid further ambiguity, we have clarified in Section 2.3.2 ("Model training and evaluation"), in the Results where we discuss Text S4 and Fig. S7, and in Supplementary Text S4, that (i) the main performance metrics are based on standard stratified k-fold CV across individual measurements, and (ii) the grouped-by-Site and grouped-by-Year CV are presented as additional diagnostic tests of transferability and sampling limitations rather than as the primary validation procedure.

**Changes in text:** Lines 290 – 293; added 3 lines after line 471

**Minor comment :**

**RC:** I'm not sure Figure's titles have been changed ? perhaps a bug ?

**AC:** We have changed the capture accordingly.

**RC:** RC: A more detailed discussion of site representativeness would be beneficial, including the number of sites per land cover type, and the number of measurements per site. In Figure 4, n appears to refer to the total number ofmeasurements, but it would also be useful to indicate the number of sites per land cover category there, as well as in Tables B1 and B2, or somewhere else. This would facilitate discussion of this limitation; e.g., the text mentions that the 'wetland, permanent' class includes only one site which.

AC: The total number of measurements per land-cover class is already shown at the top of Figure 4. To improve transparency, we have now added the number of sites per class directly in the figure caption and included both site and measurement counts in Table B1.

--> I don't think the changes worked in the figure caption.

**AC:.** We have changed Figure 4, and it's the capture accordingly. Now it includes a number of sites as well.

**RC:** RC: I do not understand Figure 7A. According to the caption, it should show monthly estimates averaged over the

entire area, but the large number of points is confusing.

AC: Figure 7A indeed shows monthly CH4 flux estimates for individual pixels across the study area, not a single

aggregated mean. The large number of points reflects spatial variability within the domain. We clarified this in the caption and indicate that each point corresponds to a pixel-level monthly mean to improve readability.

--> I think then "averaged over the entire area of interest" should be deleted.

**AC:.** We have changed capture for Fig 7 accordingly.